# Simulations of anthropogenic bromoform indicate high emissions at the coast of East Asia

Josefine Maas[1,*], Susann Tegtmeier[1,2], Yue Jia[1,2], Birgit Quack[1], Jonathan V. Durgadoo[1,3] and Arne Biastoch[1,3]

[1]GEOMAR Helmholtz Centre for Ocean Research Kiel, Kiel, Germany
[2]Institute of Space and Atmospheric Studies, University of Saskatchewan, Saskatoon, Canada
[3]Kiel University, Kiel, Germany
*now at: Helmholtz-Zentrum Geesthacht, Institute of Coastal Research, Geesthacht, Germany

*Correspondence to*: Susann Tegtmeier (susann.tegtmeier@usask.ca)

**Abstract.** Bromoform is the major by-product from chlorination of cooling water in coastal power plants. The number of power plants in East and Southeast Asian economies has increased rapidly exceeding mean global growth. Bottom-up estimates of bromoform emissions based on few measurements appear to under-represent the industrial sources of bromoform from East Asia. Using oceanic Lagrangian analyses, we assess the amount of bromoform produced from power plant cooling water treatment in East and Southeast Asia. The spread of bromoform is simulated as passive
particles that are advected using the 3-dimensional velocity fields over the years 2005-2006 from the high-resolution NEMO-ORCA0083 ocean general circulation model. Simulations are run for three scenarios with varying initial bromoform concentrations based on the range of bromoform measurements in cooling water discharge. Comparing the modelled anthropogenic bromoform to in-situ observations in the surface ocean and atmosphere, the two lower scenarios show the best agreement suggesting initial bromoform concentrations in cooling water to be around 20-60 µg L$^{-1}$. Based
on these two scenarios, the model produces elevated bromoform in coastal waters of East Asia with average concentrations of 23 and 68 pmol L$^{-1}$ and maximum values in the Yellow, Japan and East China Seas. The industrially produced bromoform is quickly emitted into the atmosphere with average air-sea flux of 3.1 and 9.1 nmol m$^{-2}$ h$^{-1}$, respectively.

Atmospheric abundances of anthropogenic bromoform are derived from simulations with the Lagrangian particle
dispersion model FLEXPART based on ERA-Interim wind fields in 2016. In the marine boundary layer of East Asia, the FLEXPART simulations show mean anthropogenic bromoform mixing ratios of 0.4–1.3 ppt, which are 2–6 times larger compared to the climatological bromoform estimate. During boreal winter, the simulations show that some part of the anthropogenic bromoform is transported by the northeasterly winter monsoon towards the tropical regions, whereas during boreal summer anthropogenic bromoform is confined to the northern hemisphere subtropics. Convective events
in the tropics entrain an additional 0.04-0.05 ppt of anthropogenic bromoform into the stratosphere averaged over tropical Southeast Asia. In our simulations, only about 10 % of anthropogenic bromoform is outgassed from power plants located in the tropics south of 20° N, so that only a small fraction of the anthropogenic bromoform reaches the stratosphere.

We conclude that bromoform from cooling water treatment in East Asia is a significant source of atmospheric bromine and might be responsible for annual emissions of 100–300 Mmol Br in this region. These anthropogenic bromoform
sources from industrial water treatment might be a missing factor in global flux estimates of organic bromine. While the

current emissions of industrial bromoform provides a significant contribution to regional tropospheric budgets, it provides only a minor contribution to the stratospheric bromine budget of 0.24-0.30 ppt Br.

## 1 Introduction

Power plants require cooling water to regulate the temperature in the system. As their demand for cooling water is very high, power plants are often located at the coast to profit from an unlimited water supply. Seawater, however, needs to be disinfected to prevent biofouling and to control pathogens in effluents. The usual disinfection method, chlorination, is known to generate a broad suite of disinfection by-products (DBPs) including trihalomethanes, halogenated acetic acids and bromate (e.g. Helz et al., 1984; Jenner et al., 1997). The generally proposed mechanism for generating DBPs is the reaction of oxidants such as chlorine and ozone with organic and inorganic substances, such as bromide (Br-) and iodide (I-), in the water via the formation of hypobromous (HOBr) and hypoiodous (HOI) acid (Allonier et al., 1999; Khalanski and Jenner, 2012). Discharge of DBPs within the cooling water effluent can be harmful to the local ecosystem in combination with temperature and pressure gradients (Taylor, 2006). The composition and amount of generated DBPs depend on many factors including the type and concentration of the injected oxidant and the chemical characteristics of the treated water such as salinity, temperature and amount of dissolved organic matter (Liu et al., 2015). Cooling water effluents regularly involve the discharge of large water volumes into the marine environment (Khalanski and Jenner, 2012). This water is often warmer than the surrounding waters and its decreased density means it stays at the sea surface. Chemicals such as DBPs contained in cooling water are likely to spread laterally across the sea surface which facilitates air-sea gas exchange for volatile DBPs.

One of the major DBPs is bromoform ($CHBr_3$), a halogenated volatile organic compound. Bromoform is also naturally produced in the ocean by macroalgae and phytoplankton and is the largest source of organic bromine to the atmosphere (Quack and Wallace, 2003). Current estimates of bromoform emissions show large variations and suggest a global contribution to atmospheric bromine (Br) of 0.5–3.3 Gmol Br a$^{-1}$ (Engel and Rigby, 2018). Bottom-up bromoform emission estimates based on statistical gap filling of observational surface data suggest in general smaller global fluxes when compared to other approaches. The bottom-up approach from Ziska et al. (2013) uses surface ocean and atmosphere measurement collected in the HalOcAt (Halocarbons in the Ocean and Atmosphere) database (https://halocat.geomar.de/) to estimate bromoform emissions of 1.5 Gmol Br a$^{-1}$. Based on physical and biogeochemical characteristics of the ocean and atmosphere, the data are classified into 21 regions and extrapolated to a regular grid within each region. Top-down bromoform emission estimates, on the other hand, are based on global model simulations adjusted to match available aircraft observations (e.g. Butler et al., 2007; Liang et al., 2010; Ordóñez et al., 2012). They are in general, a factor of two larger than bottom-up emission estimates. Individual ship cruises, aircraft campaigns and modelling studies have demonstrated a large spatio-temporal variability of bromoform in surface water and air (e.g. Fiehn et al., 2017; Fuhlbrügge et al., 2016; Jia et al., 2019). These pronounced variations combined with the poor temporal and spatial data coverage is a major challenge when deriving reliable emission estimates and may explain the large deviations between bottom-up and top-down estimates. Geographical regions with poor data coverage might not be well-represented in the global emission scenarios. Furthermore, the anthropogenic input of bromoform might be under-estimated for large industrial regions (Boudjellaba et al., 2016).

With an atmospheric lifetime of about 2–3 weeks, bromoform belongs to the so-called very short-lived substances (VSLSs) (Engel and Rigby, 2018). Once bromoform is photochemically destroyed in the atmosphere, it can deplete ozone by catalytic cycles (Saiz-Lopez and von Glasow, 2012) or change the oxidising capacity of the atmosphere by shifting $HO_x$ ratios towards OH (Sherwen et al., 2016). In the tropics, VSLSs such as bromoform can be entrained into the stratosphere (e.g. Aschmann et al., 2009; Liang et al., 2010; Tegtmeier et al., 2015) and contribute to stratospheric ozone depletion (Hossaini et al., 2015). Stratospheric entrainment of trace gases with very short lifetimes is most efficient in regions of strong, high reaching convective activity such as the Western Pacific and Maritime Continent (e.g., Pisso et al., 2010; Marandino et al., 2013). The Asian summer monsoon represents another important pathway to the lower stratosphere (e.g., Randel et al. 2010) entraining mostly Southeast Asian planetary boundary layer air. The monsoon also has the potential to include VSLSs emitted from the Indian Ocean and Bay of Bengal (Fiehn et al., 2017, 2018b). Model simulations suggest that the monsoon circulation transports the oceanic emissions towards India and the Bay of Bengal, from where they are convectively lifted and reach stratospheric levels in the south-eastern part of the Asian monsoon anticyclone. The stratospheric bromine injections from the tropical Indian Ocean and Western Pacific depend critically on the seasonality and spatial distribution of the emissions (Fiehn et al., 2018a). Model studies based on bottom-up emission estimates indicate global bromoform maxima over India, the Bay of Bengal, and the Arabian Sea as well as over the Maritime Continent and Western Pacific (Tegtmeier et al., 2020a). While aircraft measurements in the Western Pacific have confirmed high concentrations of bromoform (Wales et al., 2018), the role of the Asian monsoon as an entrainment mechanism for VSLSs has not been confirmed yet due to the lack of observations in this region.

Quantifying the contribution of bromoform to tropospheric and stratospheric bromine budgets requires reliable emission estimates that include natural and anthropogenic sources. Industrially produced bromoform will spread in the marine environment once the treated water is released and will be emitted into the atmosphere together with naturally produced bromoform. Atmospheric and oceanic measurements cannot distinguish between naturally and industrially produced bromoform and all the top-down and bottom-up emission estimates discussed above potentially include the latter already. A first comparison of natural and industrial bromoform sources from Quack and Wallace (2003) concluded a negligible global contribution of 3 % man-made bromoform. Their estimate was based on measurements of bromoform in disinfected water (80 nmol $L^{-1}$) from European power plants and cooling water use and projections of the global electricity production. In the meantime, the global electricity production has increased by almost 50 % from 16700 TWh in 2003 (IEA, 2005) to 25000 TWh in 2016 (IEA, 2018). Furthermore, new measurements of bromoform in disinfected cooling water have become available suggesting potentially higher concentrations of up to 500 nmol $L^{-1}$ (Padhi et al., 2012; Rajamohan et al., 2007; Yang, 2001). Emerging economies in East Asia, such as China and India have experienced a massive growth over the last years exceeding the global economic growth. As the existing estimate of industrially produced bromoform is outdated, updated estimates taking into account new measurements are required to assess the impact of anthropogenic activities on the production and release of brominated VSLSs as well as their contribution to stratospheric ozone depletion.

We will derive a new bottom-up VSLS emission estimate for East and Southeast Asia by quantifying anthropogenic contributions to bromoform production. We will use available cooling water measurements to predict oceanic and

atmospheric bromoform concentrations in regions of extensive industrial activities. Based on comparisons to available ocean surface and atmosphere measurements, we will evaluate our predictions and discuss implications for an atmospheric bromine budget as well as future research needs. As 50 % of the global coastal cooling water is produced in East and Southeast Asia, we define these areas as our study region. We identify locations of high industrial activity along the coast of East and Southeast Asia and derive estimates of released cooling water and therein contained bromoform (Section 2). Based on Lagrangian simulations in the ocean, we derive the general marine distribution of non-volatile DBPs released with cooling water. For the case study of bromoform, we show oceanic distributions of the volatile DBP by taking air-sea exchange into account (Section 3). Based on the oceanic emissions, the atmospheric distribution of bromoform generated in industrial cooling water is simulated with a Lagrangian particle dispersion model (Section 4). Results are compared to existing observational atmospheric and oceanic distributions (Section 5). Methods are described in Section 2, while discussion and summary are provided in Section 6.

## 2 Methods

### 2.1 Estimation of DBP production in cooling water from East Asian power plants

In this study, we investigate the oceanic distribution of DBPs produced in power plants that chlorinate seawater. We assume that all power plants located at the coast use seawater for cooling purposes. Most of the seawater is only used once in the system as the ocean provides an unlimited water supply. For the estimation of the cooling water volumes, we use the global power plant database Enipedia (enipedia.tudelft.nl, last access: 2017) where over 21,000 power plants are given together with location, electricity generation (in MWh) and sometimes fuel type. Based on the coordinates, we choose those power plants that are located less than 0.02 degrees (maximum 2 km at the equator) away from any coastline and refer to them as coastal. Based on this classification, 23 % of energy capacity from listed power plants in the database is generated by coastal power plants. The Key World Energy Statistics (IEA, 2018) give a total global electricity production of 24973 TWh in 2016. The average water use per MWh energy was given by Taylor (2006) to be 144 $m^3$ $MWh^{-1}$, which leads to a global cooling water discharge of about 800 billion $m^3$ $a^{-1}$ along the coast in 2016. For the individual coastal power plants in East and Southeast Asia, annual cooling water volumes are shown in Figure 1.

To determine the amount of bromoform produced in the cooling water, there are only a few measurements available and the locations are limited (Table 1). Most data originate from several power plants in Europe (Allonier et al., 1999; Boudjellaba et al., 2016; Jenner et al., 1997) and some studies are based on measurements from single power plants in Asia (Padhi et al., 2012; Rajamohan et al., 2007; Yang, 2001). Only Yang (2001) provides DBP measurements in East Asia. Furthermore, the location where water is sampled is not consistent among the different studies. Some samples were taken in the coastal surface water at the power plant outlet (Fogelqvist and Krysell, 1991; Yang, 2001), while other studies sampled directly inside the power plant before dilution with the ocean (Jenner et al., 1997; Rajamohan et al., 2007). The measurements show a very large variability ranging from 8–290 µg $L^{-1}$. As there is no systematic difference between measurements inside the power plant and at the power plant outlet, both types of measurements are given in Table 1 together in the first column.

In addition to the sampling location, differences in the concentrations can result from water temperature, salinity and dissolved organic carbon content, which vary with season. Colder water from mid to high latitudes during winter requires

less water treatment as the growth of pathogens takes longer compared to warm tropical or subtropical waters. The
chlorination dosage and frequency of treatment also play a distinct role for the resulting DBP concentrations (Joint
Research Council, 2001).

Given that available measurements are sparse and depend on many factors, the uncertainties in initial bromoform
concentrations in cooling water are relatively high. For our analyses we chose to scale the bromoform discharge
according to three scenarios (LOW, MODERATE and HIGH), which reflect the range of values given in available
literature (Table 1). For our simulations, we use initial bromoform concentrations of 20 µg L$^{-1}$ (LOW), 60 µg L$^{-1}$
(MODERATE) and 100 µg L$^{-1}$ (HIGH) in undiluted cooling water.

## 2.2 Lagrangian simulations in the ocean

To assess the long-term, large-scale effect of DBPs from power plant cooling water on the environment, we simulate the
distribution of non-volatile DBPs and the concentration and emission of the volatile DBP bromoform in the ocean. The
Lagrangian model runs are based on velocity output from the high-resolution, eddy-rich ocean general circulation model
(OGCM) NEMO-ORCA version 3.6 (Madec, 2008). The ORCA0083 configuration (The DRAKKAR Group, 2007) has
a horizontal resolution of 1/12 degree at 75 vertical levels and output is given at a temporal resolution of five days for
the time period 1963–2012. Atmospheric forcing comes from the DFS5.2 data set (Dussin et al., 2016). The experiment
ORCA0083-N06 used in this study was run by the National Oceanography Centre, Southampton, UK. Further details
can be found in Moat et al. (2016).

We simulate the spread of the DBPs from treated cooling water, by applying a Lagrangian trajectory integration scheme
to the 3D velocity fields with the Ariane software (Blanke et al., 1999). We perform offline trajectory calculations by
passively advecting virtual particles, which represent the DBP amount discharged with the cooling water. The calculation
of trajectories with Ariane is primarily based on advection. For each scenario we perform one simulation over the same
time period from 2005-2006. The year chosen is the same as in Maas et al. (2019), where it is shown that interannual
variability of surface velocity in the study region is small compared to seasonal variability. In each simulation, particles
are continuously released close to the power plant locations at 5-day time steps over two years. We allow for an
accumulation period of 11 months and show the results of the seasonal and annual mean of the second year starting in
December 2005. A detailed description of the applied method can be found in Maas et al. (2019).

Our study focusses on the region of East and Southeast Asia (90° E–165° E, 10° S–45° N), which comprises 50 % of the
global coastal power plant capacity and cooling water discharge. The particle discharge locations have been chosen as
close to the coastlines as possible (Figure 2). Particles are released approximately 8 to 40 km offshore, as the model-
resolution does not allow to capture smaller-scale coastal structures such as harbours or estuaries nor does it simulate the
near-costal exchange, e.g. through tides. Our approach ensures minimal influence of the land boundaries on the
simulation in order to avoid numerically-related beaching of particles into the coastal boundary.

We conduct two different simulations allowing us to analyse the spread of long-lived DBPs in general and the spread of
bromoform as a specific case. First, we simulate the spread of a passive tracer, which does not have any environmental
sinks and is representative of long-lived non-volatile DBP. We consider the full history of simulated particle positions,

which is equivalent to assuming no particles getting lost through sinks in the ocean or emission into the atmosphere. The resulting distribution shows locations where non-volatile DBPs such as bromoacetic acid are transported through the ocean currents within one year.

Second, we simulate the spread of bromoform as a major volatile DBP including the simulation of atmospheric fluxes and oceanic sinks. Each particle is assigned an initial mass of bromoform according to the amount of cooling water used by the respective power plant (Figure 1) and the bromoform concentration prescribed by the three scenarios, MODERATE, HIGH and LOW. The particle density distribution is calculated at the sea surface down to 20 m on a $1° \times 1°$ grid. The distribution is given as particle density per grid box in percent for non-volatile DBPs and as concentration in pmol $L^{-1}$ for bromoform.

For the second set of simulations, the sink processes of bromoform such as constant gas exchange at the air-sea interface or chemical loss rates are taken into account. The air-sea flux of bromoform is calculated following the general flux equation at the air-sea interface:

$$\text{Flux} = (C_w - C_{eq}) \cdot k \qquad (1)$$

Here Flux is positive when it is directed from the ocean to the atmosphere and is given in pmol $m^{-2}$ $h^{-1}$. $C_w$ is the actual concentration in the surface mixed layer in pmol $L^{-1}$ and

$$C_{eq} = C_{air} \cdot H_{CHBr3}^{-1} \qquad (2)$$

is the theoretical equilibrium concentration at the sea surface (in pmol $L^{-1}$) calculated from the atmospheric mixing ratio (in ppt) and the Henry's Law constant $H_{CHBr3}$ of bromoform. The gas transfer velocity k (in cm $h^{-1}$) mainly depends on the surface wind speed and temperature and is calculated following Nightingale et al., (2000). Wind velocities at 10 m height are taken from the NEMO-ORCA forcing data set DFS5.2 (Dussin et al., 2016), which is based on the ERA-Interim atmospheric data product.

As the oceanic and atmospheric terms in the air-sea flux parameterisation are of additive nature, it is possible to calculate the flux of anthropogenic and natural bromoform separately. For our simulations, we only consider bromoform from cooling water and apply the air-sea flux parameterisation to the anthropogenic portion of bromoform in water and air. We have conducted sensitivity tests (see section 2.3) to estimate the impact that atmospheric bromoform abundances have upon the flux calculations. The tests show that outgassed anthropogenic bromoform leads to atmospheric surface values $C_{eq}$, which are always below 8 % of the underlying sea surface concentration $C_w$ (at a water temperature of 20°C). Such low equilibrium concentrations can be considered negligible for the flux calculation and therefore $C_{eq}$ is set to zero in our study.

The sea surface concentration and air-sea flux from the three simulations are also compared to climatological maps of bromoform concentration and emissions from the updated Ziska et al. (2013) inventory (hereafter referred to as Ziska2013) (Fiehn et al., 2018a).

Mean sea surface concentrations $C_w$ are calculated by averaging over the area where 90 % of all released bromoform accumulates. To this end, all grid cells are sorted according to descending bromoform concentrations and the average is calculated over the first grid cells that contain in total 90% of all bromoform. Maximum concentrations are calculated by averaging over the area where 10 % of the highest bromoform values accumulate. Mean and maximum air-sea fluxes

are calculated using the same averaging principle as for $C_w$. The annual mean atmospheric bromine flux resulting from industrial bromoform emissions in East and Southeast Asia is derived from the air-sea flux maps of the whole domain.

## 2.3 Lagrangian simulation in the atmosphere

Based on the seasonal mean emission maps, we obtain a source function of atmospheric bromoform. We simulate the atmospheric transport and distribution of bromoform for the three different emission strength scenarios with the Lagrangian particle dispersion model FLEXPART (Stohl et al., 2005). The FLEXPART model includes parameterisation for moist convection (Forster et al., 2007) and turbulence in the boundary layer and free troposphere (Stohl and Thomson, 1999). It has been used in previous studies with a similar model setup and shown robust VSLS profiles compared to observations (e.g. Fiehn et al., 2017; Fuhlbrügge et al., 2016; Tegtmeier et al., 2020a).
Seasonal mean bromoform emissions derived from the three scenarios are used as input data at the air-sea interface over the East and Southeast Asia area defined as our study region. The meteorological input data (temperature, and winds) stem from the ERA-Interim reanalysis product (Dee et al., 2011) and are given on a 1°×1° horizontal grid, at 61 vertical model levels and a 3-hourly temporal resolution. The chemical decay of bromoform in the atmosphere was accounted for by prescribing a lifetime of 17 days during all runs (Montzka and Reimann, 2010).
The FLEXPART simulations were performed for boreal winter (December–February, DJF) and summer (June–August, JJA) seasons, respectively, each with a two-month spin-up phase. Since there are only weak dynamical variations between different years, we used an ensemble mean of four years (2015-2018) each. A total of 1000 particles are randomly seeded inside each grid box at each time step according to the air-sea flux strength. Output mixing ratios are given at the same horizontal resolution and 33 vertical levels from 50 to 20000 m. Detailed descriptions of model settings are described in Jia et al. (2019).

We perform three additional FLEXPART runs, Ziska2013-EastAsia, Ziska2013 and Ziska2013+MODERATE based on the updated Ziska2013 emission inventory with the same FLEXPART configuration as described above for both seasons, DJF and JJA. As the Ziska2013 inventory currently presents our best knowledge of bottom-up bromoform emissions, it is of interest to analyse how much of these emissions can be explained by industrial sources and how much stems from natural sources.
The Ziska2013-EastAsia run uses only the Ziska2013 climatological emissions over East and Southeast Asia defined as our study region. Results from Ziska2013-EastAsia in the atmospheric boundary layer are used to compare the mixing ratios based on our anthropogenic emissions in the East and Southeast Asia region.
For comparisons of mixing ratios in the free troposphere and upper troposphere/lower stratosphere (UTLS), air-sea fluxes from other parts of the tropics also need to be taken into account as the time scales for horizontal transport are often shorter than the ones for vertical transport. Therefore, we set up the runs, Ziska2013 and Ziska2013+MODERATE. Ziska2013 uses the air-sea flux of the Ziska2013 climatology for the global tropics and subtropics between 45° S and 45° N. As the Ziska2013 climatology is taking into account only very few northern hemispheric coastal data points, it likely neglects anthropogenic fluxes in some regions. Therefore, the Ziska2013+MODERATE run uses the Ziska2013 fluxes between 45° S and 45° N, but replaces them with the anthropogenic MODERATE flux values in all grid boxes where the MODERATE fluxes are larger than the Ziska2013 fluxes. The two runs, Ziska2013+MODERATE and

Ziska2013, are used to quantify the additional anthropogenic bromoform based on the MODERATE scenario in the UTLS region. The UTLS region is calculated as the height of the cold point tropopause, which has been derived from ERA-Interim model level data at 6 hourly resolution (Tegtmeier et al., 2020b).

Mean mixing ratios from the whole domain (90° E–165° E, 10° S–45° N) in the marine boundary layer and in the UTLS
are given as the average over the 90 % area characterised by the highest local values, and maximum mixing ratios as the average over the largest 10 % (see Section 2.2). In a second step, we define two regions in order to analyse the vertical transport of bromoform into the free troposphere and into the UTLS. For the height profiles of the Ziska2013 and the Ziska2013+MODERATE runs, we average mixing ratios over a region above the Maritime Continent, which we refer to as the tropical box (10° S–20° N, 90° E–120° E), and over another region from China to Japan which we refer to this as
the subtropical box (30° N–40° N, 120° E–145° E) (Figure 2).

## 3 Oceanic spread of DBPs and bromoform

The particle density distribution shows the annual mean DBP accumulation pattern in the region of interest in East and Southeast Asia (Figure 3). Non-volatile DBPs from cooling water usually accumulate around the coast and in the marginal seas. There is a clear latitudinal gradient with only little DBP distribution south of 20° N, except for higher
values in the Strait of Malacca. In contrast to the relatively low DBP density in the inner tropics, the subtropics show a very high accumulation of DPBs with a centre in the marginal seas between 25° N and 40° N. While power plants can be found along all coastlines (Figure 1), the power plant capacity and therefore the amount of treated cooling water is much higher along the subtropical coasts of China, Korea and Japan leading to the DBP distribution pattern shown in Figure 3. Hot spots are around the coast of Shanghai and Incheon with a DBP density of 1 %. A relatively high DBP
density of 0.8 % can also be found in the East China Sea, the Yellow Sea, the southern Japan Sea, the Gulf of Tonkin and the Strait of Malacca. Medium to low DBP density in the South China Sea suggests only small contributions of cooling waters to this region. Since Japan and Korea have a large number of power plants with high volumes of cooling water discharge, a relatively large amount of DBPs is transported eastward with the Kuroshio Current east of Japan into the North Pacific.

The distribution of bromoform, as a volatile DBP in the surface ocean differs from the DBP accumulation pattern shown in Figure 3, because the volatile DBPs are outgassed into the atmosphere. The annual mean sea surface concentration of bromoform from cooling water is shown in Figure 4 (panel a-c) for the three cooling water discharge scenarios LOW, MODERATE and HIGH and with a substantially smaller spread compared to the non-volatile DBPs. The area which
contains the 90 % highest bromoform concentrations does not vary between the three scenarios, as the air-sea flux, which determines how much bromoform remains in the water, is linearly proportional to the sea surface concentration. Higher surface concentrations result in higher fluxes into the atmosphere, which limits the spread of bromoform substantially compared to non-volatile DBPs. Bromoform concentrations are around 23, 68, and 113 pmol $L^{-1}$ (LOW, MODERATE and HIGH) averaged over the region where the 90 % of bromoform with the highest concentrations accumulate (Table
2). This region is to a large degree limited to latitudes north of 20° N as a result of the power plant distribution. As in the case of the non-volatile DBPs, most of the bromoform is centred along the Chinese, Korean and Japanese coast line with

a larger spread into the marginal seas for the latter two. One exception to this latitudinal gradient is the Strait of Malacca where local power plants result in average bromoform concentrations of 3.4, 10.3 and 16.7 pmol L$^{-1}$ (LOW, MODERATE, and HIGH).

Observational based oceanic bromoform concentrations from Ziska2013 (Figure 4, panel d) are relatively evenly spread along the coastlines of the region and do not show the latitudinal gradient found for the anthropogenic concentrations. North of 20° N the anthropogenic bromoform is much higher than the oceanic distribution from Ziska2013, where the maximum lies around 21 pmol L$^{-1}$. Our simulations reach maximum values (averaged over the 10 % highest bromoform concentrations) of 112, 338 and up to 563 pmol L$^{-1}$ (LOW, MODERATE and HIGH, Table 2) in the Japan Sea. These
concentrations are all above 100 pmol L$^{-1}$ and are very high compared to observational values from Ziska2013 (Figure 4, panel d).

Air-sea fluxes of anthropogenic bromoform show a similar distribution as the oceanic concentrations (Figure 5, panel a-c). Flux rates averaged over the region of the 90 % highest flux values are 3, 9 and 15 nmol m$^{-2}$ h$^{-1}$ (LOW, MODERATE
and HIGH). Maximum flux rates (averaged over the highest 10 %) even reach 13, 41 and 68 nmol m$^{-2}$ h$^{-1}$ in the Japan Sea near the Korean and Japanese coast for the three scenarios (Table 2). In contrast, the existing observational based estimates from the Ziska2013 climatology peak with 1.1 nmol m$^{-2}$ h$^{-1}$ located in the South China Sea along the west coast of the Philippines (Figure 5, panel d).

The annual bromine input from the ocean into the atmosphere in the form of bromoform emissions in the East and Southeast Asia region is 118 Mmol Br according to the observation-based inventories from Ziska2013 (Table 2). Our simulations suggest that the anthropogenic input alone amounts to 100, 300 and 500 Mmol Br a$^{-1}$ (LOW, MODERATE, HIGH) for the same region, which corresponds to almost 99 % of the bromine produced during cooling water treatment in the power plant for each scenario. This implies that all bromoform from cooling water treatment is eventually
outgassed from the ocean into the atmosphere. While average and maximum air-sea fluxes of anthropogenic bromoform are much higher and confined to small areas around the discharge locations, the Ziska2013 air-sea fluxes are distributed along all coastlines and the equator and result in similar total annual mean Br flux as the LOW emission scenario (Table 2). 90 % of the annual mean atmospheric bromine input from anthropogenic bromoform in East Asia occurs north of 20° N where 89–447 Mmol Br are released over one year, compared to the tropical Southeast Asian regions south of
20° N where only 10–52 Mmol Br a$^{-1}$ enter the atmosphere (from LOW to HIGH). In contrast, only 29 % of the total bromine from the Ziska2013 climatology in East Asia is released into the atmosphere north of 20° N, which suggests that the majority of the anthropogenic emissions from this region are missing in the Ziska2013 climatology.

## 4 Anthropogenic bromoform in the atmosphere

### 4.1 Mixing ratios in the marine boundary layer

Atmospheric mixing ratios of anthropogenic bromoform are derived from FLEXPART runs driven by the seasonal emission estimates discussed in section 3. Atmospheric bromoform from industrial emissions is shown for a seasonal average in the marine boundary layer at 50 m height for JJA for all three scenarios (Figure 6, panel a-c). Mean mixing ratios are 0.4, 1.3 and 2.3 ppt (LOW, MODERATE, HIGH, Table 2). Overall, high atmospheric mixing ratios are found

around the coastlines of Japan, South Korea and northern China. Although maximum emissions are located in the Japan Sea, maximum mixing ratios are mostly located south of Japan with values up to 9.0, 27.1 and 45.0 ppt (LOW, MODERATE, HIGH, Table 2). Here, the westerlies lead to bromoform transport from the Japan Sea into the Northwest Pacific. We also localise hotspots of strong anthropogenic bromoform accumulations due to enhanced emissions over Shanghai, Singapore or the Pearl River Delta, respectively (Figure 6, panel a). During boreal summer, the Western Pacific and Maritime Continent are influenced by southwesterly winds and the anthropogenic bromoform experiences northward transport, bringing some smaller portion of the subtropical emissions into the mid-latitudes (Figure 6, panel a, b and c).

During boreal winter (DJF, Figure 7, panel a-c), anthropogenic bromoform shows somewhat lower atmospheric mixing ratios with a mean of 0.3, 0.8 and 1.4 ppt and maximum values of 4.7, 13.5 and 23.3 ppt for the three scenarios (Table 2). In contrast to boreal summer, the atmospheric transport is dominated by winds from the northeast and higher bromoform values are confined to tropical and subtropical regions (Figure 7). Thus, tropical mixing ratios show a clear seasonal variability and are on average over 3 times higher for DJF than for JJA without large shifts in the location of the bromoform emissions (Figure S1).

In order to compare the atmospheric impact of industrial emissions with existing results, we repeat our analysis for the bottom-up emissions scenario Ziska2013 for the same region, which has been frequently used in past studies (e.g. Hossaini et al., 2013, 2016). Atmospheric mixing ratios are derived from seasonal FLEXPART runs driven by Ziska2013-EastAsia and shown for a seasonal average at 50 m height for JJA (Figure 6d). For both seasons, JJA and DJF, atmospheric bromoform based on industrial emissions is larger than atmospheric bromoform based on the Ziska2013-EastAsia emissions (Figure 6d, Figure 7d). These differences are maximised in the subtropical regions, where anthropogenic bromoform dominates especially during JJA when anthropogenic mixing ratios are 2–5 times larger compared to climatological Ziska2013 bromoform (for LOW and MODERATE). In the tropical regions, the situation is more complicated. Atmospheric abundances driven by the industrial emissions reach higher peak values of up to 2 ppt especially in the Strait of Malacca (MODERATE, Figure 6b), while mixing ratios driven by the observationally based emissions from Ziska2013-EastAsia are smaller only reaching peak values of up to 0.8 ppt, but are spread over a much wider area (Figure 6d). Given the comparison of the boundary layer values, it is not clear, which emission scenario will result in a larger contribution to the stratospheric halogen budget.

## 4.2 Mixing ratios in the free troposphere and UTLS

In order to analyse atmospheric transport from the marine boundary layer into the free troposphere and UTLS, seasonal mean bromoform mixing rations are averaged over a subtropical box (30° N–40° N, 120° E–145° E, Figure 2) and a tropical box (10° S–20° N, 90° E–120° E, Figure 2) from the Ziska2013 and Ziska2013+MODERATE simulations for DJF and JJA. Both simulations are based on global climatological Ziska2013 bromoform air-sea fluxes between 45° S and 45° N, with Ziska2013+MODERATE including additional anthropogenic bromoform fluxes in East and Southeast Asia.
In the subtropical box (Figure 8), there is a strong dominance of anthropogenic bromoform in the marine boundary layer during JJA several times higher compared to bromoform from climatological bottom-up emissions (Figure 8). Our

simulations suggest that during convective events in JJA, anthropogenic bromoform from the subtropical marine boundary layer can be transported into the UTLS region, up to the height of the cold point. In our simulation Ziska2013+MODERATE, convective events during the summer bring on average over 0.3 ppt into the UTLS (Figure 8). During DJF (not shown), there is only very little transport of bromoform out of the boundary layer, and entrainment of
anthropogenic bromoform into the subtropical UTLS is confined to boreal summer when the intertropical convergence zone (ITCZ) is located north of 10° N (Waliser and Gautier, 1993).

In the tropical box (Figure 9), atmospheric bromoform mixing ratios in the marine boundary layer are generally weaker than in the subtropics (Figure 8). The seasonal difference between DJF and JJA is only pronounced in the tropical marine
boundary layer for Ziska2013+MODERATE, where mixing ratios during DJF exceed 0.5 ppt throughout the whole time period (Figure 9a) and are around 0.4 ppt during JJA (Figure 9b). The air-sea fluxes show no strong seasonal variations, therefore this difference must be transport-driven. During DJF, the prevailing northeasterly winds advect the bromoform from regions of high anthropogenic emissions in East Asia towards the Maritime Continent, increasing the tropical bromoform abundance substantially. Thus, tropical convection during DJF can transport more of the anthropogenic
bromoform emitted in East Asia into the UTLS compared to similar events during JJA (Figure 9). The difference between the Ziska2013+MODERATE and the Ziska2013 average mixing ratios in the UTLS is 0.05 ppt during DJF and 0.04 ppt during JJA. These values present the anthropogenic contribution to stratospheric bromine from East and Southeast Asian cooling water based on the MODERATE bromoform emission scenario.

Atmospheric processes over the Maritime Continent, which encloses the tropical box, are characterised by deep convective events, that can lead to entrainment of VSLSs into the stratosphere (Aschmann and Sinnhuber, 2013; Tegtmeier et al., 2020a). For our case study, convective events reaching the UTLS occur in both seasons (Figure 9). Moreover, here is a clear anthropogenic signal from Ziska2013+MODERATE compared to Ziska2013 in the free troposphere in this region in both seasons, which is more pronounced during DJF (Figure 9a) than during JJA (Figure
9b) in agreement with the elevated mixing ratios in the marine boundary layer.

In addition to the mixing ratios averaged over two boxes, we show the spatial distribution of seasonally averaged bromoform mixing ratios at the cold point tropopause for the whole domain (90° E–165° E, 10° S–45° N) (Figure 10) based on the Ziska2013 and Ziska2013+MODERATE emissions. During DJF (Figure 10a and c), there is a clear
anthropogenic signal over the Bay of Bengal, across the equator towards Indonesia. Mixing ratios for the Ziska2013+MODERATE run are 0.22±0.07 ppt averaged over the area of 90 % highest mixing ratios and 0.18±0.05 ppt for Ziska2013, which implies that 0.04 ppt is of anthropogenic origin (Table S1). Again, the strong advective transport in the boundary layer during DJF bringing higher bromoform abundances from the subtropics into the tropics plays an important role here. Some fraction of the advected bromoform is picked up by tropical deep convection and transported
into the UTLS and up to the cold point. As the latter represents the stratospheric injection level, the interaction of boundary layer advection and local convection over the Indian Ocean and Maritime Continent results in an efficient transport pathway for anthropogenic bromoform from industrial sources in East Asia to the stratosphere.
During JJA (Figure 10b and d), mean bromoform mixing ratios averaged over the area of 90 % highest mixing ratios are slightly smaller, with 0.20±0.06 ppt and 0.15±0.05 ppt based on the Ziska2013+MODERATE and Ziska2013 emissions,

respectively (Table S1). During the Asian summer monsoon, the region of main upward transport of VSLS lies at about 20°N over the Indian Ocean so that the main stratospheric injection region of VSLSs shifts to the Bay of Bengal and northern India (Fiehn et al., 2018b). However, most of the boundary layer bromoform from anthropogenic sources stays in the northern hemisphere around the coastline of China and over the Western Pacific thus decoupled from the monsoon convection.

While over 90 % of anthropogenic bromoform is outgassed north of 20° N, our simulations show that the additional anthropogenic emissions in the MODERATE scenario contribute on average 0.05 ppt $CHBr_3$ during JJA to 0.04 ppt $CHBr_3$ during DJF to the stratospheric bromine budget at the UTLS (Table S1). This is an increase of 22-32 % compared to the Ziska2013 mixing ratios of 0.15 ppt and 0.18 ppt $CHBr_3$ during JJA and DJF, respectively.

## 5 Discussion

### 5.1 Comparison with bromoform measurements in the ocean

Observations of bromoform in the surface ocean and atmosphere from East and Southeast Asia can help to determine which scenario (LOW, MODERATE, HIGH) offers the best fit for simulating anthropogenic bromoform in this region. Recent measurement campaigns show elevated bromoform concentrations in the coastal waters of the East China and Yellow Seas (He et al., 2013a, 2013b; Yang et al., 2014, Yang et al., 2015). Average values of 6–13 pmol $L^{-1}$ were
measured in the Yellow and East China Seas during boreal spring and summer (Yang et al., 2014; Yang et al., 2015), and 17 pmol $L^{-1}$ were measured in boreal winter (He et al., 2013b). Particularly high concentrations were detected by He et al. (2013a) during spring in the East China Sea with a mean of 134 pmol $L^{-1}$. Highest bromoform concentration over 34 pmol $L^{-1}$ (He et al., 2013b) and over 200 pmol $L^{-1}$ (He et al. 2013a) were observed near the estuaries of the Yangtse River, which the authors attributed to anthropogenic activities including coastal water treatment in the Shanghai region.
Our simulations also show mean surface concentrations around Shanghai of 14–71 pmol $L^{-1}$ (LOW to HIGH), in the range of the observations by He et al. (2013a).
Measurements in the South China and Sulu Seas (Fuhlbrügge et al., 2016) show a high variability of bromoform in the surface seawater with average concentrations of 19.9 pmol $L^{-1}$. Highest values of up to 136.9 pmol $L^{-1}$ are found close to the Malaysian Peninsula and especially in the Singapore Strait suggesting industrial contributions. Maximum
anthropogenic bromoform from our simulations in the Singapore Strait ranges from 36–178 (LOW to HIGH), in good agreement with maximum values reported by Fuhlbrügge et al., (2016).
Average anthropogenic bromoform concentrations for the three scenarios are around 23–113 pmol $L^{-1}$ (averaged over the region of the 90 % highest values, Table 2) and are larger than the observational average values. The larger model values might be due to the fact that the cooling water effluents do not distribute far into the marginal seas but stay near the coast
as observed by Yang (2001) and confirmed by our simulations. Our simulated anthropogenic bromoform concentrations stay usually within 100 km of the coast, the averaged observational values, however, include also measurements that are up to 200 km away from the coastline and can therefore be expected to be lower. While observational mean values are slightly lower than our model results, maximum values found close to the coast line show very good agreement with the model results.

## 5.2 Comparison with bromoform measurements in the troposphere

An extensive study of atmospheric measurements over South Korea and adjacent seas was performed in spring (May and June) 2016 from the Korea-United Sates Air Quality Study (KORUS-AQ; https://www-air.larc.nasa.gov/missions/korus-aq/). The aircraft measurements of various VSLSs including bromoform were repeatedly taken between 0 and 12 km in the region between 30° N–40° N and 120° E–145° E coinciding with our subtropical box discussed earlier (Figure S2). The data used here, is based on the 60 second merged dataset from all flight sections. In the campaign region around South Korea, an average bromoform atmospheric mixing ratio from all sections of 2.5±1.4 ppt was measured in the lower 100 m (Figure 8). In comparison, our simulations for the Ziska2013+MODERATE scenario show an average mixing ratio of 3.8±1.4 ppt in the lowest 100 m in the subtropical box during JJA. The simulations based on Ziska2013 give a bromoform mixing ratio of only 0.3±0.1 ppt for the same altitude range demonstrating that the additional anthropogenic bromoform results in a much better agreement with the observations in the marine boundary layer around South Korea.

Above the boundary layer, mixing ratios from KORUS-AQ rapidly decline to 0.5–0.7 ppt in the 3–9 km altitude range (Figure 8). Here, the Ziska2013+MODERATE simulation suggest seasonal mean mixing ratios between 0.4–0.7 ppt, which fit very well to the KORUS-AQ data. Simulations based on Ziska2013 suggest 0.2 ppt bromoform in this region clearly underestimating the observations (Figure 8). Between 9 and 12 km, the observed bromoform values drop sharply to values around 0.2±0.08 ppt suggesting that the airplane probed air masses above the convective outflow. The smooth seasonal mean profiles from the two simulations do not show such a sharp decrease of values and in consequence the lower Ziska2013 results agree better with the observations in the height 9–12 km. In general, the comparison with the KORUS-AQ data shows that our simulations agree quite well the observations in the middle troposphere when anthropogenic emissions from cooling water treatment in East Asia are included based on the MODERATE scenario.

Additional bromoform mixing ratios from observations near large industrial cities are shown in Table 3. In the subtropical East China Sea, surface measurements are available and atmospheric mixing ratios of 0.9 ppt and 0.3 ppt were found during boreal winter and summer, respectively (Yokouchi et al., 2017). Our simulations in the East China Sea suggest anthropogenic bromoform contributions of 1.7–5.1 ppt near Shanghai, being on the upper side of the observations (Table 3). Nadzir et al. (2014) observed relatively high values in the South China Sea (1.5 ppt) during boreal summer. Our simulations show average mixing ratios of 0.5–1.8 ppt at the surface (LOW to MODERATE) near the Pearl River Delta in the South China Sea, in good agreement with Nadzir et al. (2014) (Table 3). Around Singapore, high oceanic bromoform concentrations were measured of 4.4 ppt during JJA (Nadzir et al., 2014) and 3.4 ppt during DJF (Fuhlbrügge et al., 2016). Our simulations result in peak bromoform mixing ratios near Singapore of 1.4–4.3 ppt for JJA and of 1.7–5.3 ppt for DJF (LOW and MODERATE), respectively, in good agreement with Nadzir et al. (2014) and Fuhlbrügge et al. (2016) (Table 3). Especially the high atmospheric bromoform mixing ratios found near Singapore and the Pearl River Delta can be associated with anthropogenic activity.

The HIGH scenario shows average mixing ratios, which are in general too high for the whole domain. Thus, it is not likely that cooling water treatment produces anthropogenic bromoform with average concentrations of 100 µg L$^{-1}$. Nevertheless, such concentrations can occur at some locations and produce extremely high bromoform abundances near the coast of industrial regions, as confirmed by the observations presented here.

## 5.4 Discussion of uncertainties

Our analyses suggests that anthropogenic bromoform accumulates in the boundary layer increasing the bromine budget in East and Southeast Asia by 85-254 % compared to the Ziska2013 climatology. This input can be expected to impact tropospheric bromine budget and ozone chemistry. While we have not analysed these aspects in our study, it should be investigated in follow-on projects. The highest uncertainties in the estimates presented here, arise from the highly variable bromoform amounts found in chemically treated cooling water. Since there are very few and no recent measurements from power plants in East and Southeast Asia available, the chosen scenarios aim to give a range of environmental concentrations of anthropogenic bromoform. Additional uncertainties can arise from oceanic and atmospheric transport simulations and the parameterisation of air-sea fluxes. Since bromoform is emitted into the atmosphere on very short timescales, uncertainties arising from oceanic transport simulations are small compared to scenario uncertainties. Similarly, given the high saturation of anthropogenic bromoform in surface water, the sensitivity of our results to the air-sea flux parameterisation can be expected to be small.

Atmospheric modelling can introduce additional uncertainties, especially regarding the contribution of anthropogenic sources to stratospheric bromine. VSLS FLEXPART simulations have been evaluated in numerous previous studies and shown in most cases to give good agreement with upper air observations (e.g., Fuhlbruegge et al., 2016; Tegtmeier et al., 2020a). In summary, uncertainties of our results are dominated by uncertainties of the bromoform concentrations in undiluted cooling water. We have successfully reduced these uncertainties by nearly a factor of two based on comparing our predictions to available observations.

## 5.4 Discussion of stratospheric entrainment

If bromoform is entrained into the stratosphere, it will contribute to ozone depletion driven by catalytic cycles. Atmospheric transport simulations show that during boreal winter strong northeasterly winds transport the anthropogenic bromoform from the East China Sea towards the tropics. Here, it can be taken up by deep convection and reach the cold point tropopause thus being entrained into the stratosphere. On average, 0.22 ppt of bromoform are entrained above the cold point based on natural and additional anthropogenic emissions (from the MODERATE scenario). For the same configuration during boreal summer, the large amount of anthropogenic bromoform emitted over the East China Sea do not reach the tropics, resulting in average mixing ratios of 0.20 ppt at the cold point level. In summary, the high anthropogenic bromoform emissions in the East China, Yellow and Japan Seas do not efficiently reach the stratosphere, except for some fraction that is advected with the Asian winter monsoon into the tropics, in which case it can lead to an increased entrainment of 22–32 % over this area when compared to Ziska2013. Comparison with measurements up to 12 km in the subtropics shows that the simulated bromoform agrees very well with the observations if additional anthropogenic sources from the MODERATE scenario are included. The good agreement suggests that anthropogenic bromoform can lead to an additional stratospheric entrainment of 0.04–0.05 ppt $CHBr_3$, which corresponds to a bromine input of 0.12–0.15 ppt Br.

This study focusses on source gas entrainment into the stratosphere and does not take into account additional product gas entrainment resulting from anthropogenic bromoform sources. Most observational and modelling studies estimate the total stratospheric bromine contribution to be split half and half into source and product gas contributions (Engel and

Rigby, 2018 and references therein). Therefore, we estimate the total stratospheric bromine contribution in the form of both, source gas and product gases, from the East and Southeast Asia anthropogenic bromoform sources to be around 0.24–0.30 ppt Br. Compared to a total stratospheric bromine contribution from all VSLSs of about 3–7 ppt Br (Engel and Rigby, 2018), the anthropogenic input estimated in this study provides only a minor contribution.

## 6 Summary and conclusions

We predict that there is a strong anthropogenic source of bromoform along the coast of East Asia with particular large contributions north of 20° N from the East China, Yellow and Japan Seas. This anthropogenic source results from local cooling water treatment in power plants and leads to extremely high annual mean air-sea flux rates of 3.1–9.1 nmol m$^{-2}$ h$^{-1}$ in coastal waters in East Asia. Simulations of atmospheric bromoform originating from industrial sources show an accumulation in the marine boundary layer and result in mean bromoform mixing ratios of 0.4–1.3 ppt. The simulations show a strong seasonal variability with high bromoform abundances being transported into the mid-latitudes during boreal summer and into the tropics during boreal winter. In comparison, the bottom-up inventory by Ziska2013 shows much lower concentrations along the coast of East Asia, but higher mean sea surface concentrations in Southeast Asia. Our predictions are based on assuming initial bromoform concentrations in chemically treated cooling water from power plants. These concentrations depend on many different factors and observational studies provide a range of 8–290 μg CHBr$_3$ L$^{-1}$. We take the large range of possible bromoform concentrations into account by analysing three different scenarios that assume LOW, MODERATE and HIGH bromoform concentrations in undiluted cooling water.

We evaluate our predictions by comparing the model results to available measurement data in the ocean and atmosphere. Comparisons with some individual campaigns suggest that our averaged anthropogenic values based on the MODERATE scenario agree very well with the observations. For other campaigns, the model results overestimate campaign-averaged bromoform concentrations in surface water and air. The latter discrepancy of the mean values is possibly related to the regional extent of the specific campaign data, given the very sharp bromoform gradients from the coast into the open ocean waters. Maximum values found in surface water and air during the campaigns, however, agree very well with our estimates based on industrial sources for the LOW and MODERATE scenarios in nearly all cases. Oceanic and atmospheric abundances based on the HIGH scenario are likely too high and only results based on the two lower scenarios are presented in this summary.

Our predictions and their evaluation indicate that cooling water from power plants provides a substantial source of anthropogenic bromoform. Depending on the scenario, 100 to 300 Mmol Br a$^{-1}$ are released into the atmosphere from the coastal regions in Southeast and East Asia (LOW to MODERATE) in the form of anthropogenic bromoform. The largest part, about 90 %, are emitted in coastal regions north of 20° N. In comparison, the Ziska2013 climatology estimates bromoform emissions of 34 Mmol Br a$^{-1}$ for the same region north of 20° N. The high emissions of industrially produced bromoform in East Asia are most likely underrepresented in existing bottom-up estimates by Ziska2013 and Stemmler et al. (2015) in these regions and might explain some of their differences when compared to top-down estimates. However, the additional input from anthropogenic bromoform sources in Southeast Asia makes only a minor contribution to stratospheric bromine. Our models show that the anthropogenic bromoform emissions from their major

sources in the East China, Yellow and Japan Seas do not efficiently reach the stratosphere. The anthropogenic contribution from product and source gases to stratospheric bromine is estimated to be around 0.24–0.30 ppt Br, which is small compared to the total estimate of stratospheric bromine of 3–7 ppt Br.


This study suggests that current bottom-up bromoform climatologies miss large anthropogenic sources. Further targeted measurement campaigns in coastal, shelf and open ocean regions and dedicated monitoring of DBPs at coastal sites are required to estimate the regional extent and distribution of anthropogenic bromine sources. Detailed information about the water volumes used for each power plant, as well as the disinfection technique can also help to better localise regions

of high DBP discharge.

While this study exclusively looks at the DBPs from cooling water treatment in power plants, other anthropogenic sources also contribute to local and global emissions of organic bromine, like desalination plants or ballast water from commercial ships, which produce DBPs in chemically treated water. Desalination is mostly done in the Arabian Peninsula (Jones et al., 2019) and does not play a large role in Southeast Asia. Ballast water volumes of 3–5 billion $m^3$ $a^{-1}$

(Tamelander et al., 2010) are globally negligible compared to cooling water volumes from coastal power plants but can locally increase DBP discharge (Maas et al., 2019). For assessing the total impact of anthropogenic VSLSs on a local industrial area, such as Singapore or the Pearl River Delta region, all sources of chemical water treatment need to be taken into account. Direct outgassing during treatment of circulating water through the cooling towers into the atmosphere can also occur, which has not been quantified yet and is therefore not considered here. Overall, cooling water

from power plants can be assumed to be the largest global source of anthropogenic bromoform as it has by far the largest water volumes and is present in all regions and climate zones. The contribution of bromoform from anthropogenic sources should be considered as relevant next to natural sources for future estimates of the global bromine fluxes.

**Acknowledgements**

The OGCM model data used for this study were kindly provided through collaboration within the DRAKKAR framework by the National Oceanographic Centre, Southampton, UK. We especially thank Andrew C. Coward, Adrian L. New and colleagues for making the data available. The OGCM and trajectory simulations were performed in the High-Performance Computing Centre at the Christian-Albrechts-Universität zu Kiel. Furthermore, we wish to thank Bruno Blanke and Nicolas Grima for realising and providing the Lagrangian software ARIANE; and Siren Rühs for helping

with the set-up of the ARIANE environment. And many thanks to Donald Blake for providing the KORUS-AQ bromoform measurements. This study was carried out within the Emmy-Noether group AVeSH (A new threat to the stratospheric ozone layer from Anthropogenic Very Short-lived Halocarbons) funded by the Deutsche Forschungsgemeinschaft (DFG, German Research Foundation) – TE 1134/1. JVD acknowledges the Helmholtz-Gemeinschaft and the GEOMAR Helmholtz Centre for Ocean Research Kiel (grant IV014/GH018).

**Author contribution**

JM wrote the manuscript, performed the Lagrangian ocean simulations and created the output. YJ performed the Lagrangian simulations in the atmosphere. ST developed the research question and guided the research process. BQ

developed the research question and gave input on the observational data. AB and JVD provided the NEMO-ORCA model data and gave input on the ocean simulations. All authors took part in the process of the manuscript preparation.


The authors declare that they have no conflict of interest.

**Data availability**

Data from the ARIANE and FLEXPART simulations are available upon request from the corresponding author. The KORUS-AQ data are available from https://www-air.larc.nasa.gov/cgi-bin/ArcView/korusaq?MERGE=1.

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

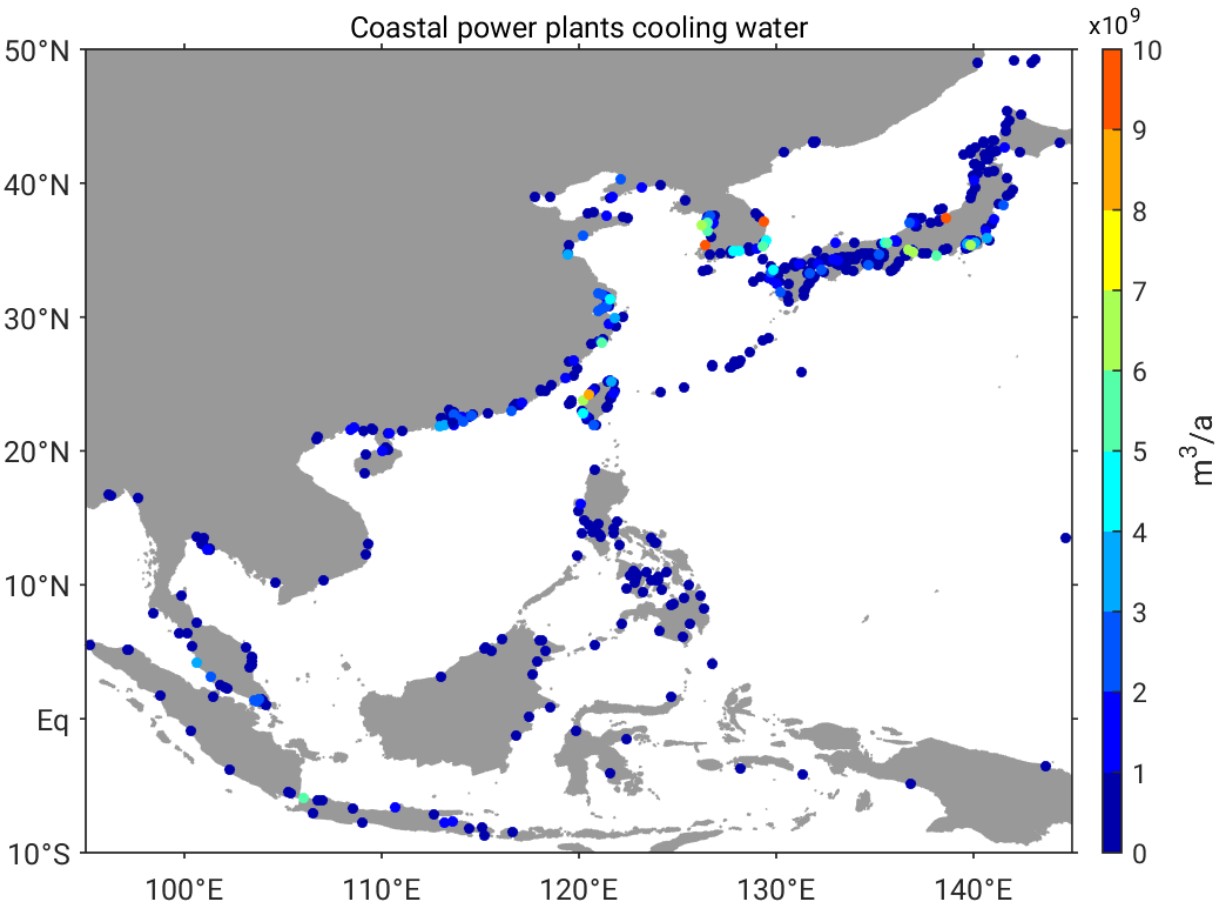


Figure 1: Location and annual cooling water volume [billion $m^3$ $a^{-1}$] of coastal power plants in East and Southeast Asia extracted from the enipedia-database and color-coded by the cooling water discharge.

Table 1: Bromoform concentrations measured in water samples from power plant cooling water and surrounding waters. Measurements in the power plant effluent can refer to both, samples of the undiluted water stream or sea water samples at the outlet.

| Power plant effluent/ near outlet | | Surroundings | | Location | Reference |
|---|---|---|---|---|---|
| µg L$^{-1}$ | nmol L$^{-1}$ | µg L$^{-1}$ | nmol L$^{-1}$ | | |
| 90-100 | 356-396 | 1-20 | 4-79 | Gothenburg, Sweden, Kattegatt | Fogelqvist, 1991 |
| 9-17 | 35-67 | 0.1-5 | 0.4-20 | North Sea | Jenner, 1997 |
| 8-27 | 32-107 | n/a | n/a | English Channel | Allonier, 1999 |
| 124 | 495 | 1-50 | 4-200 | Youngkwang, South Korea, Yellow Sea | Yang, 2001 |
| 20-290 | 79-1147 | 0-54 | 0-214 | Kalpakkam, India, Bay of Bengal | Rajamohan, 2007 |
| 12-41 | 47-162 | n/a | n/a | Kalpakkam, India, Bay of Bengal | Padhi, 2012 |
| 19 | 75 | 0.5-2.2 | 2-9 | Gulf of Fos, France, Mediterranean | Boudjellaba, 2016 |

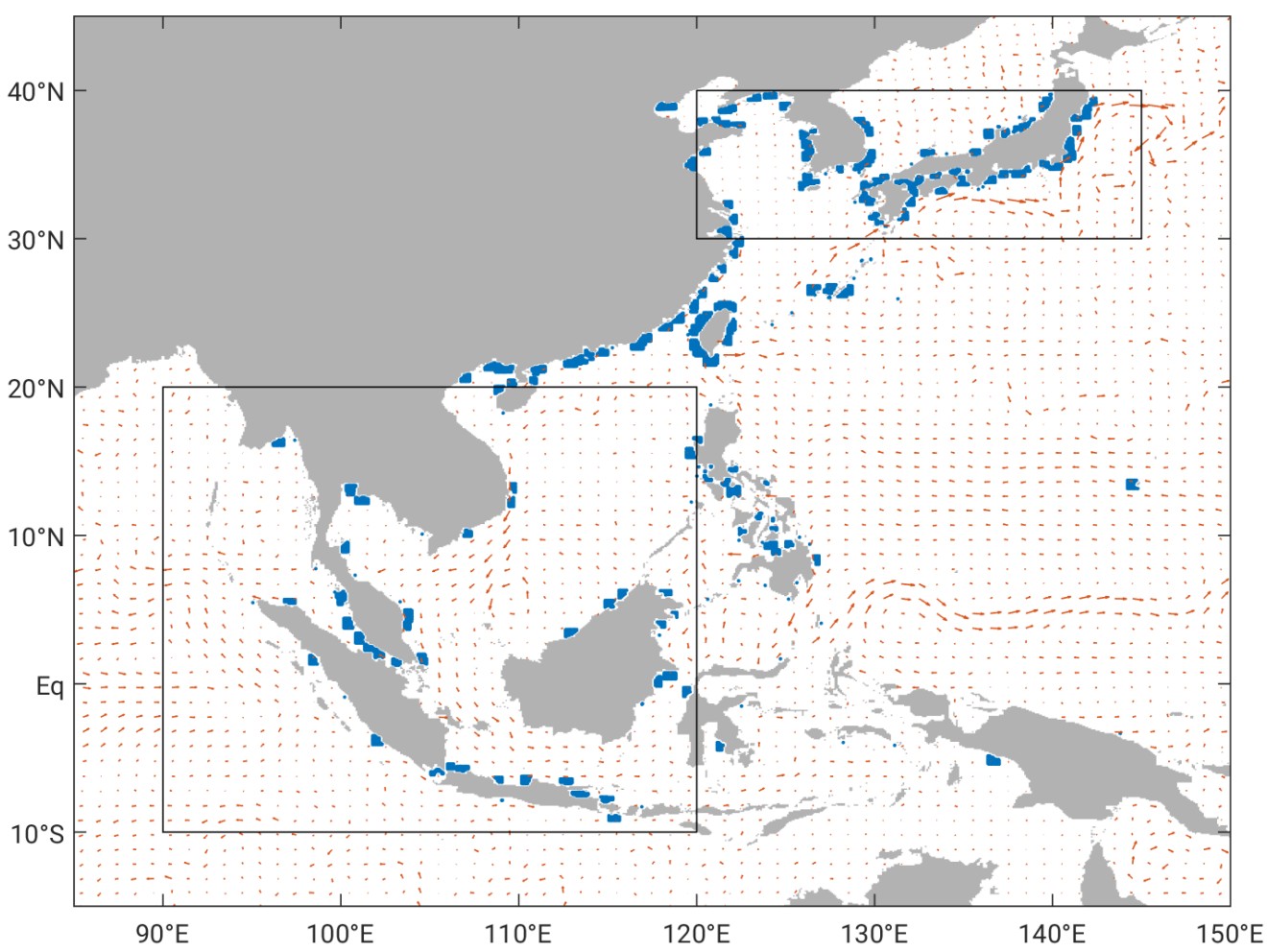

Figure 2: Initial position of particles in East and Southeast Asia (blue dots). NEMO-ORCA12 ocean currents from the initialisation time in January 2005 (red arrows); and the two boxes, which mark the region referred to as tropics and subtropics as described in section 2.3.

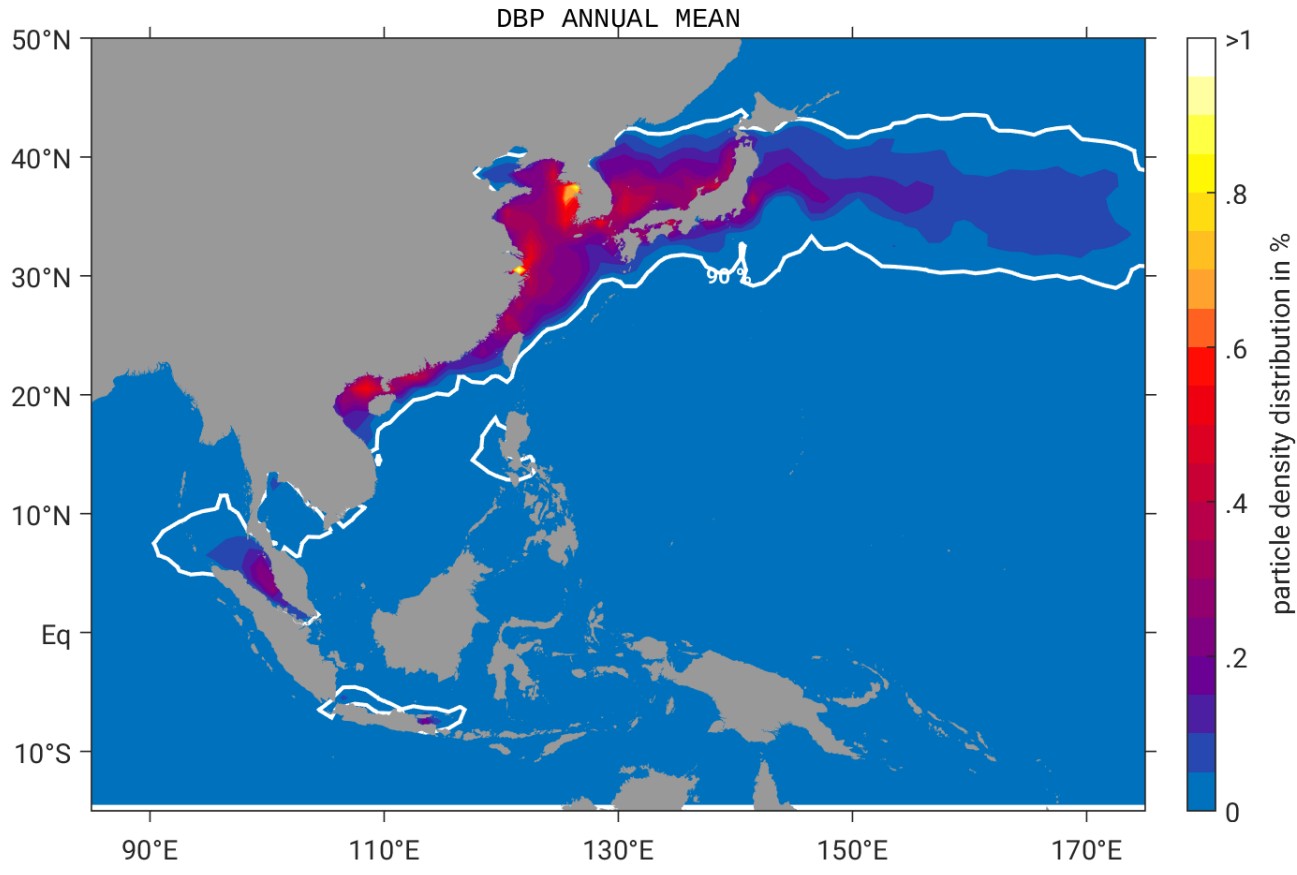

Figure 3: Annual mean particle density distribution in % of DBPs from cooling water treatment in coastal power plants in East and Southeast Asia. The white contour line shows the patches where 90 % of the largest particle density are located.

Table 2: Mean and maximum bromoform concentration [pmol L$^{-1}$] for the three Ariane runs LOW, MODERATE, HIGH, as well as the climatological values from the Ziska2013 bottom-up estimate in East and Southeast Asia. Mean and maximum air-sea flux [pmol m$^{-2}$ h$^{-1}$] from the three scenarios and the Ziska2013 air-sea flux in East and Southeast Asia. The annual mean bromine flux [Mmol Br a$^{-1}$] is derived from the air-sea flux of the total domain in East and Southeast Asia. Mean and maximum atmospheric bromoform mixing ratios in the marine boundary layer [ppt] from the four FLEXPART runs. Values are given as the mean and the standard deviation averaged over the largest 90 % (referred to as mean values) and over the largest 10 % (referred to as maximum values).

| Scenario | Sea surface concentration [pmol L$^{-1}$] | | Air-sea flux [pmol m$^{-2}$ h$^{-1}$] | | Bromine flux [Mmol Br a$^{-1}$] | Atmospheric CHBr$_3$ mixing ratio [ppt] | | | |
|---|---|---|---|---|---|---|---|---|---|
| | | | | | | JJA | | DJF | |
| | Mean | Max | Mean | Max | Total | Mean | Max | Mean | Max |
| LOW | 23 ± 24 | 112.1 ± 6.3 | 3.1 ± 3.4 | 13.7 ± 0.9 | 100 | 0.4 ± 0.9 | 9.0 ± 1.3 | 0.3 ± 0.5 | 4.7 ± 2.6 |
| MODERATE | 68 ± 74 | 338.3 ± 16.6 | 9.1 ± 10.2 | 41.1 ± 2.9 | 300 | 1.3 ± 2.7 | 27.1 ± 3.5 | 0.8 ± 1.4 | 13.5 ± 7.7 |
| HIGH | 113 ± 122 | 563.6 ± 28.8 | 15.1 ± 16.9 | 68.5 ± 4.7 | 500 | 2.2 ± 4.4 | 45.0 ± 6.3 | 1.4 ± 2.4 | 23.3 ± 12.9 |
| Ziska2013-EastAsia | 7 ± 6 | 21.3 ± 1.3 | 0.4 ± 0.2 | 1.1 ± 0.2 | 118 | 0.2 ± 0.1 | 0.8 ± 0.2 | 0.2 ± 0.1 | 0.5 ± 0.1 |

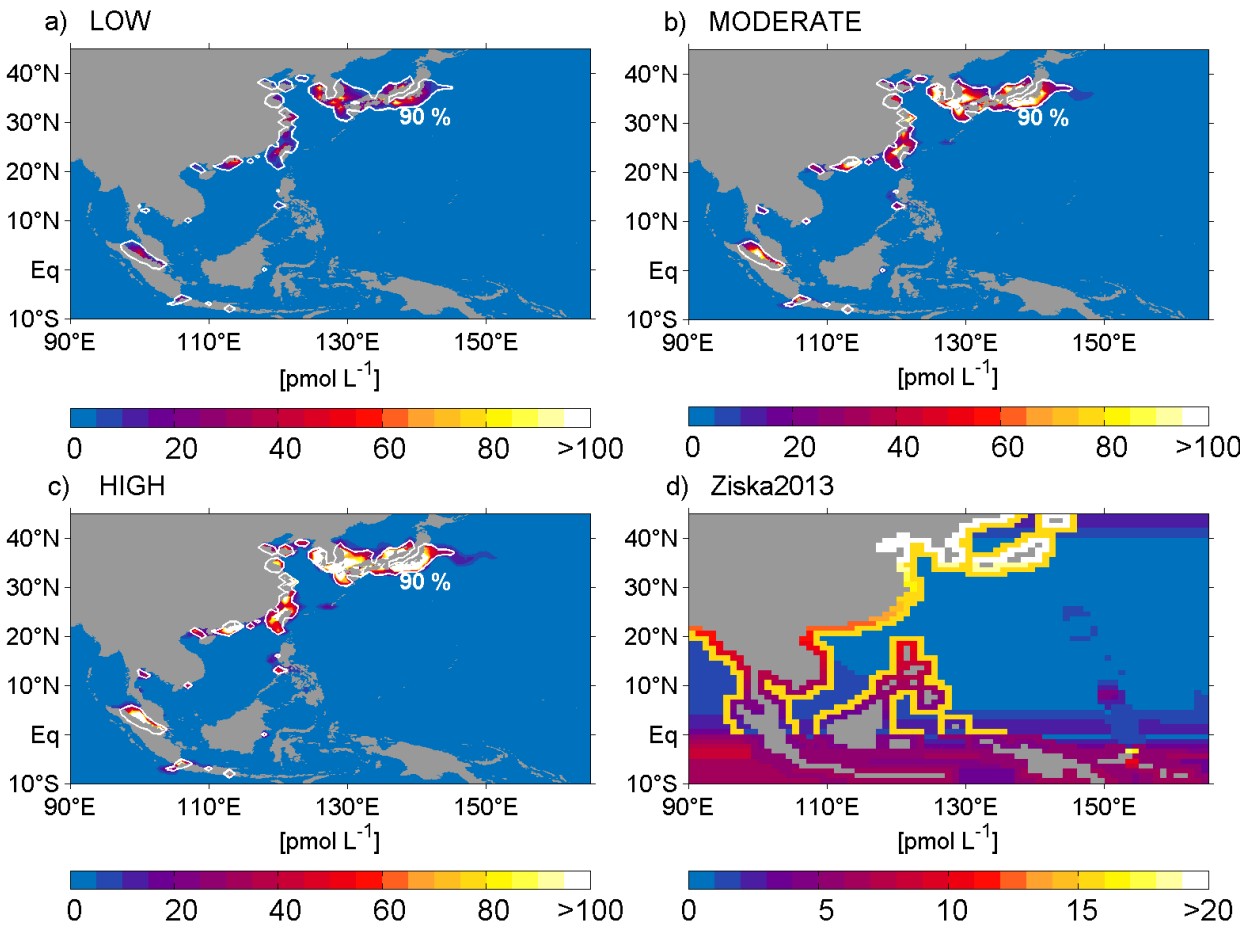

Figure 4: Annual mean surface bromoform concentration in pmol L$^{-1}$ for the three scenarios a) LOW, b) MODERATE and c) HIGH as well as d) the bromoform surface map updated from Ziska2013. Note, that the colourbar limits for d) varies from the limits in a)-c). The white contour line in panel a)-c) shows the patches where 90 % of the largest concentrations are located.

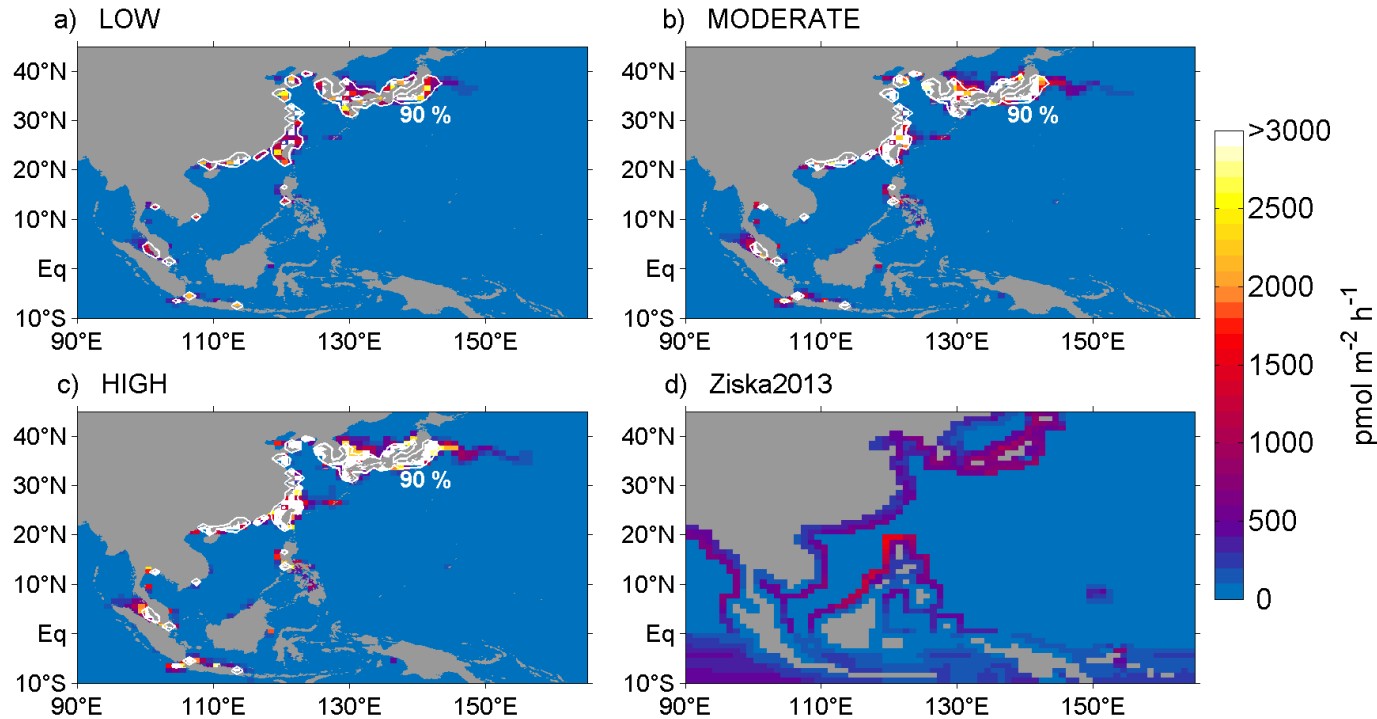

Figure 5: Annual mean air-sea flux of bromoform in pmol m$^{-2}$ h$^{-1}$ for the three scenarios a) LOW, b) MODERATE, c) HIGH, as well as d) the air-sea flux calculated from updated ocean and atmospheric maps following Ziska2013. The white contour line in panel a)-c) shows the patches where 90 % of the largest emissions are located.

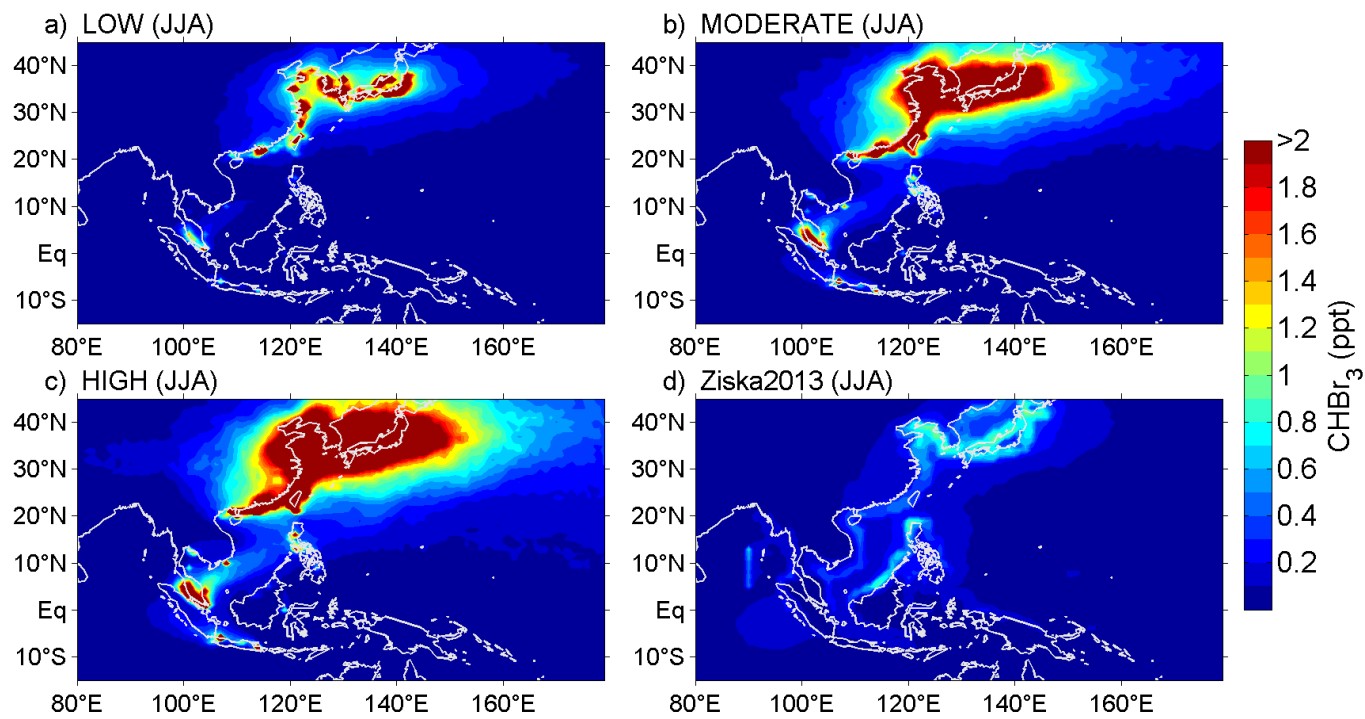

Figure 6: Seasonal mean bromoform mixing ratios [ppt] in 50 m height during JJA derived from FLEXPART runs driven by the three scenarios a) LOW, b) MODERATE, c) HIGH, and d) Ziska2013-EastAsia.

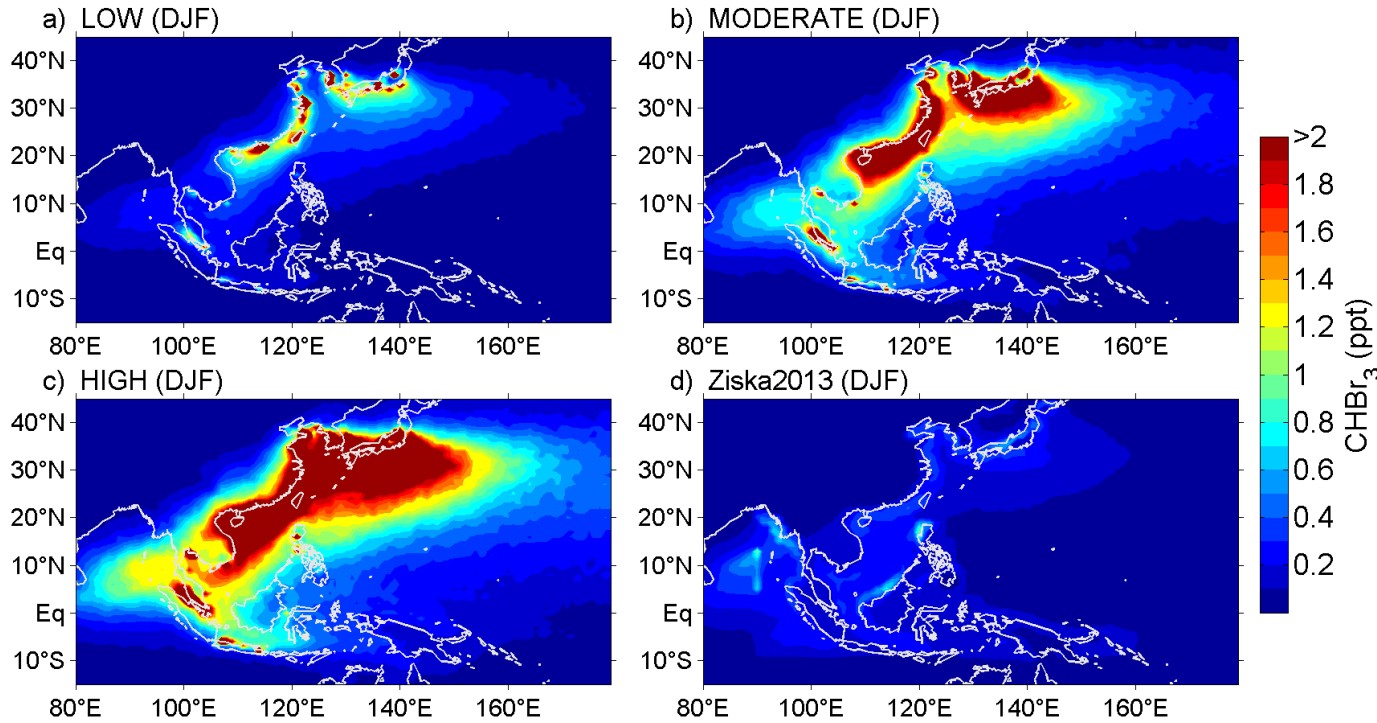

Figure 7: Same as Figure 6 only during DJF.

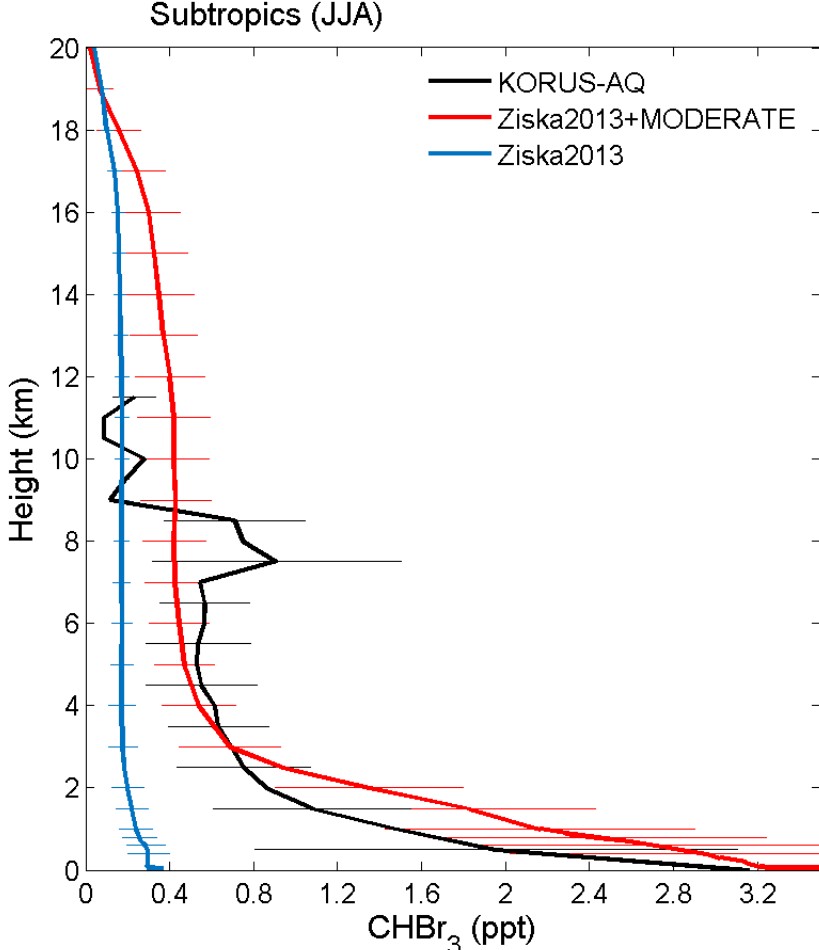

Figure 8: Height profile of seasonal mean bromoform mixing ratio [ppt] in the subtropics (30° N–40° N, 120° E–145° E) during JJA for the Ziska2013+MODERATE run (red) and the Ziska2013 run (blue). Additionally shown is the averaged profile of bromoform measurements from the KORUS-AQ campaign over South Korea and the Yellow Sea (black). Horizontal lines show the standard deviation for specific heights.

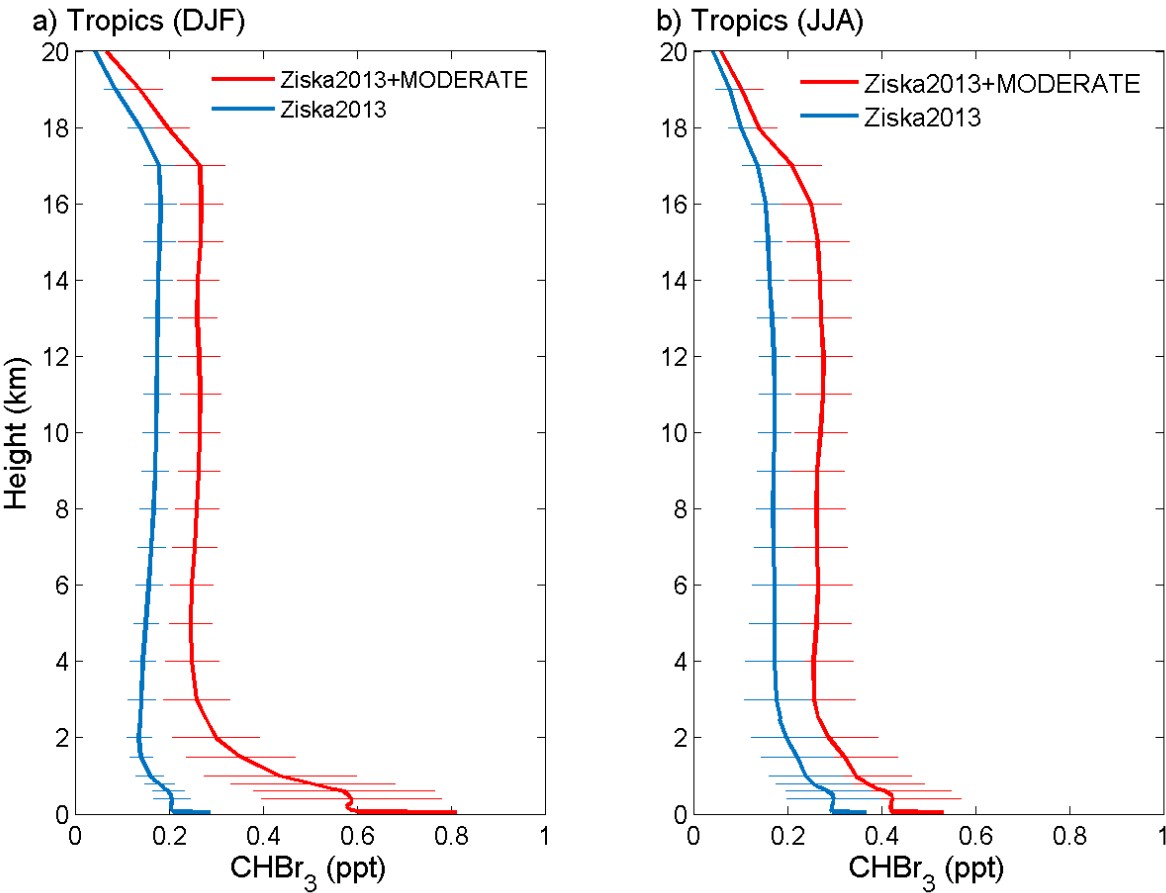

Figure 9: Height profile of seasonal mean bromoform mixing ratio [ppt] in the tropics (10° S–20° N, 90° E–120° E) for the Ziska2013+MODERATE run (red) and Ziska2013 run (blue) for both a) DJF and b) JJA. Horizontal lines show the standard deviation for specific heights.

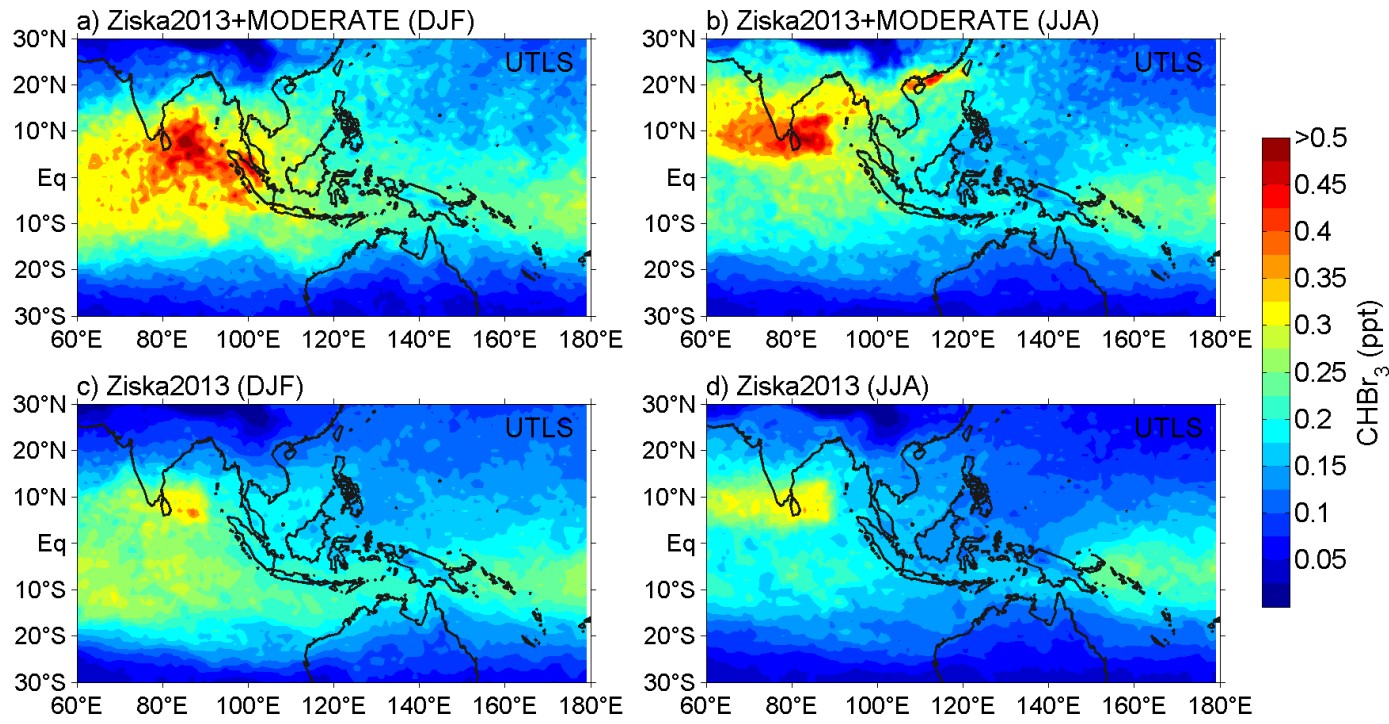

Figure 10: Seasonal mean atmospheric CHBr$_3$ mixing ratios [ppt] for a) and b) the Ziska2013+MODERATE and c) and d) Ziska2013 simulation at the cold-point tropopause height for DJF (left) and JJA (right).

Table 3: Observations with reference and corresponding range of mean modelled bromform mixing ratios [ppt] from the FLEXPART simulations LOW and MODERATE from this study.

| Region | Season | CHBr$_3$ [ppt] | Reference |
| --- | --- | --- | --- |
| Subtropical East China Sea | DJF | 0.9 | Yokouchi et al. (2017) |
| Subtropical East China Sea | JJA | 0.3 | Yokouchi et al. (2017) |
| South China Sea | JJA | 1.5 | Nadzir et al. (2014) |
| Singapore | JJA | 4.4 | Nadzir et al. (2014) |
| Singapore | DJF | 3.4 | Fuhlbrügge et al. (2016) |
| Shanghai (East China Sea) | DJF | 1.7-5.0 | This Study |
| Shanghai (East China Sea) | JJA | 1.8-5.1 | This Study |
| Pearl River Delta (South China Sea) | DJF | 1.0-3.0 | This Study |
| Pearl River Delta (South China Sea) | JJA | 0.5-1.8 | This Study |
| Singapore | DJF | 1.7-5.3 | This Study |
| Singapore | JJA | 1.4-4.3 | This Study |