# Peer review of "Simulations of anthropogenic bromoform indicate high emissions at the coast of East Asia"

_Atmospheric Chemistry and Physics, 2019_

## Referee Comment (RC2)

This paper describes the estimation of sea surface concentrations of bromoform over East Asia resulting from the treatment of power station coolant water with chlorine-based disinfectants. The authors subsequently then estimate sea to air fluxes and test these within an atmospheric transport model to estimate the transport of bromoform in the atmosphere, in particular, via convection to the stratosphere. The authors estimate the sea surface concentrations using a bottom-up approach by first estimating bromoform within power station coolant waters, the discharge of this coolant water into the ocean, and then its subsequent transport in the ocean using oceanic transport model. Based on the bottom-up approach, the authors find a notable contribution to oceanic bromoform from this source that strongly affects oceanic bromoform concentrations close to the source region. Based on the sea-to-air fluxes and the atmospheric modelling, despite showing significant anthropogenic bromoform levels near the source regions and in the boundary layer, the authors find only a modest contribution of this source to stratospheric bromoform mixing ratios.

Bromoform has an important relevance to stratospheric ozone depletion. This paper therefore covers an important topic since current knowledge of bromoform sources is currently highly uncertain due to only limited monitoring of this gas. Attempts to improve our knowledge of its sources are therefore highly welcome. I therefore find that the paper sits well within the scope of the journal.

The subject matter and overall concept and methodology of the paper alone make it worthy for publication. On the whole, this is a good piece of work. Despite the manuscript's strengths though, there are several weaknesses in the work. I therefore have a list of general and specific comments that will need to be addressed to improve the manuscript to a sufficient level to merit final publication.

**General Comments**

1. **The overall framing of the paper**. I think it would help ease some concerns (see for instance reviewer #1's comments) if the paper's main findings regarding anthropogenic bromoform were to be framed as a set of predictions (that can be tested) based on independent knowledge and data. Your bottom-up method makes a set of predictions based on the data you have used. Then you have made efforts to validate those predictions using sea surface bromoform concentration and atmospheric measurements; these data provide somewhat limited but promising support for your predictions. As it is, the manuscript abstract and conclusions contains various statements that are rather definitive, e.g., "We find that bromoform…" (line 26in the abstract) and, similarly, "We find that…" (line 388 conclusions). When in fact the observational support appears promising yet far from definitive, and would require more dedicated observational monitoring to really prove this hypothesis. I therefore propose the authors modify the manuscript so it clearly follows this chain of reasoning: prediction/hypothesis (emission atmospheric, oceanic modelling) into observational support, followed by evaluation, and then lastly clear statements on what is needed to provide stronger support for this hypothesis, i.e., more specific and targeted observations.

2. **Treatment/explanations of the UTLS and cold point**. I think that the UTLS and cold point definition is too simplistic. The use of UTLS throughout does not capture any of the essential details of this complex atmospheric region. Sticking to a single altitude of 17 km for the cold point is also not really realistic when looking at different latitudes and seasons. Furthermore, sticking to the cold point as a definition of the lower stratosphere alone is also not entirely suitable and the suggestion from reviewer #1 to simply use the 380-390 K potential temperature line also misses some of the subtle complexity. I recommend that the authors consult Corti et al. 2005 and 2006 (see below) that both provide clear explanations and observational support for more nuanced explanations of dynamical interactions in the UTLS. First, the level of zero radiative heating (LZRH) is also useful a measure for whether air masses will undergo slow radiatively driven ascent to

above the 380-390 K levels, and it is usually at 15 km for clear sky conditions. Second, they show that in-cloud (cirrus) radiative heating can be responsible for lofting cloud containing air masses from as low as 11 km upwards to eventually reach the stratosphere. I would recommend that the authors try to calculate the LZRH using the ECMWF meteorological fields to try to diagnose this to help determine which airmasses at altitudes below 17 km are heading up or down; this would really strengthen the paper and strongly aid the interpretation of the results, which is quite difficult at this point. If this is not possible it would be very useful to see at least the 11 and 15 km levels, but this would be a much weaker alternative.

3. **Many missing details**. There are several missing important details from various sections of the paper, e.g., the year that is studied – this is not mentioned at all. I have addressed each of my concerns in specific comments below.

4. **Clarity of the manuscript**. At many points the information given is insufficient to understand precisely what is being said. I have made various specific remarks below to help address this concern.

5. **Duration of FLEXPART simulations/transport times to the stratosphere**. The duration of the FLEXPART runs is 3 months. As shown in Corti et al. 2005/2006, transport times from the boundary layer to the 390 K level of 50 days, and so even with a 1 month spin-up, we can expect a delay of ~20 days for air masses from the beginning of the spin up to reach this level. Indeed, this appears to be visible in Figs. 8b and 9c and d. This complicates the interpretation of the results both for the 5-day averages (i.e., when are they in the course of the simulations), and for the time series in Figs. 8 and 9. Similarly, the bromoform emitted in the last ~50 days of the 3 month simulations has no chance to reach these altitudes. Please discuss these issues.

6. **Discussion of key results**. Despite being an important component of the bromine lofted to the stratosphere resulting from bromoform, the authors make no mention of product gases (PGs). This is particularly important in light of comment #2 above. I do not expect the authors to simulate PG formation, chemistry, washout, and transport, but it should be clearly explained that we expect much of the bromoform to be chemically processed into PGs during the 50 days or so of vertical ascent to 390 K from the boundary layer. Further on this point, it would be worth having some discussion on CTM studies showing the partitioning of bromoform and PGs at different levels in the atmosphere.

7. **Year of study**. As mentioned, the year of study is not mentioned in the paper. While it could be interesting to do a multi-year analysis, if this is beyond the capabilities/time constraints of the authors, an alternative would be to provide some climatological context on the specific year of study. The WMO annual climate reports usually give a good region by region analysis that would help to set the meteorological context.

*References*

Corti, T.,  B. P. Luo  T. Peter  H. Vömel  Q. Fu., Mean radiative energy balance and vertical mass fluxes in the equatorial upper troposphere and lower stratosphere, Geophysical Research Letters, https://doi.org/10.1029/2004GL021889, 32 (6), 2005.

Corti, T., Luo, B. P., Fu, Q., Vömel, H., and Peter, T.: The impact of cirrus clouds on tropical troposphere-to-stratosphere transport, Atmos. Chem. Phys., 6, 2539-2547, https://doi.org/10.5194/acp-6-2539-2006, 2006.

**Specific Comments**

I found the abstract to unclear and at time contradictory from line 20 onwards. I think this stems from the fact that the authors try to say a bit too much at the same time while also only partially introducing terms such as "bottom-up estimates". Here, this is a specific reference to the prior work Ziska et al.

2013, but when I initially read this it appears to be a reference to the method in the current work since one could also classify this as a bottom-up inventory set of sea-air fluxes.

I found issue with the single number 0.03 pptv of bromoform in the stratosphere quoted in the abstract. Firstly, given that two scenarios are discussed (LOW and MODERATE), it seemed odd to only quote a single value. Further to this point, the authors look at two different seasons yet only quote one value – again, please resolve this issue. Second, if this is an average, it would make sense to quote the associated standard deviation. Third, there is no context or explanation given for this number: is it a temporal average, a spatial average, what is the duration of the average? These are all important details that would allow readers to understand the results.

Line 43. It would make sense to show some of the chemical equations associated with bromoform formation in coolant water if they are known.

The ordering of the introduction was a bit disjointed in my opinion and to also contain some text that is not relevant to the work at hand. I would remove the sentences between lines 50 and 55. In my opinion the text should be reordered such that the paragraph on lines 70-82 should be the second paragraph. The third paragraph should then be on lines 56-68. This would make a more logical flow in my opinion.

Line 117. It was not clear what you meant by the settlement of pathogens. Do you mean growth?

Line 137. I do not claim that this is important to their findings, but the authors should justify not including diffusion.

Line 140. Please mention the years you are looking at in this study and in Mass et al. 2019.

Line 149. I could not make sense of the following text. It was not clear how point 2 relates to the text that follows or where point 2 is discussed. It was unclear what "distinguish" meant in this context – this is too vague and a more precise explanation would be welcome. Are points 1 and 2 meant to describe separate simulations? Separate processes? And why are 1 and 2 being treated separately at all? Clearer explanations here would be very helpful to the clarity of the manuscript.

Line 171. The authors should make it clearer how the values of $C_{eq}$ are calculated from the outgassed bromoform; this is currently not explained.

Lines 167-173. In general, this section of text needs to be clearer. This could be improved by stating that the low $C_{eq}$ values are driven by low atmospheric vmr. It would also be clearer if the authors stated how $C_{eq}$ relates to vmr.

Line 178. "Mean concentrations are calculated…". In air, $C_W$, or $C_{eq}$ or atmospheric vmr?

Line 178-179. "…of bromoform, characterised by the highest local concentrations, accumulate." This is not very clear.

Line 180. "Mean and maximum fluxes are calculated based on the same principle." What principle?

Line 181. "The annual mean atmospheric bromine input from industrial bromoform emissions". I think you mean *resulting* instead of "input".

Section2.3. We are missing a lot of details here. What resolution are the simulations carried out on? The same resolution as the meteorology? Are the emissions constant during a season? Are Lagrangian particles emitted over the entire ocean and then the emission rate is proportional to the air-sea flux? What year are you looking at?

Line 195 onwards. We are told that there are three additional runs that are made. Then, over the course of four paragraphs with at times unclear descriptions we are told details about them, but they are only referred to as 'first run' and then 'two additional runs', and then 'first of two runs'. These descriptions are imprecise and confusing. Please can the authors define three names for the runs in

line 195 first and then describe them in the following text as "Run A does this….Run B does that … etc".

Line 214. The authors refer to means of the whole domain, but what is the domain?

Line 214-215. I could not understand the descriptions as they are for "Mean mixing ratios from the whole domain in the marine boundary layer and in the UTLS are given as the average over the 90 % area characterised by the highest local values, and maximum mixing ratios as the average over the largest 10 % (see Section 2.2)." Also, how did the authors decide upon the 90% and 10% levels?

Line 216. The authors say they identify two regions. I think they mean *define*.

Line 221. "…pattern in the research area of interest (Figure 3)." I think the authors mean region, and also which region? There are different areas being talked about. Please be precise for clarity.

Line 231. Please can the authors show the Kuroshio current on the map?

Lines 259-264. The section is unclear. The sentence on lines 262-264 is particularly unclear. Also, for clarity sake, please refer consistently to the Ziska et al. 2013 emissions as Ziska2013. These sentences are confusing because information is expressed imprecisely and there are references to prior statements that themselves unclear. Please try to arrange the information clearly, methodically, and logically.

Line 264. Is the implication of the results that most of the East Asian CHBr3 in Ziska2013 is anthropogenic in origin? I think the authors should state this more clearly if this is the prediction.

Line 266. What is the 29% percentage relative to?

Line 271. I found it odd that the authors make a 3 month long simulation and then only show a 5-day average in that entire simulation. Please can the authors explain or justify why such a short period of time is selected? Could the authors consider either monthly or 3-monthly averages as well? Also, which 5 days is this from within the 3 month simulation? All instances of this should be made clear and/or justified.

Lines 288-299. The authors discuss Figure 6 in relation to this text but do not mention the DJF results in Figure 7.

Lines 302-303. From the description given, it is not entirely clear what has been averaged. I assume it is a spatial average, but the authors should specify because the sentence implies it is spatial and temporal.

Sentence on lines 321-324. I suggest placing this sentence prior to the sentence beginning "Thus, …" on line 320.

Line 333. Please explain when and where the 5-day snapshot is.

Line 340 and 343. Please state when and where the vmr values are calculated for.

Lines 363-369. I am concerned here at the averaging approach reduces the complexity and is masking effects of over sampling of the open ocean regions. Thus, I am not sure this shows a good comparison of the same thing. I think this highlights that more thorough statistical analysis needs to be carried out, i.e., a simple x versus y spatial scatter plot. Including this would strengthen the conclusions of the paper.

Line 378. There is no mention of the year under comparison. Providing that there is overlap in the year, the KORUS-AQ data suggested by reviewer #1 could be useful here.

Line 388. Recommend changing "find" to *predict*.

Line 392. Make sure it is clear these are simulated vmrs.

Line 392. What is a cloud of high bromoform? Perhaps use something more precise like "A diffuse area with high bromoform abundances".

Line 395-396. Please be more specific as this sentence is unclear.

Line 403. Recommend stating that the assumptions are reasonable in the majority of case since the cited observations show larger ranges than those stated here.

Line 406. Recommend stating that the HIGH results are only too high in the majority of cases.

Line 408. Recommend being more specific. Instead of "results" state *bottom-up emissions, modelling, and observations*.

**Technical Comments**

Recommendations. Please use a comma after uses of which in cases where it introduces a nonrestrictive phrase. When describing using a method from another publication use *following* instead of after.

Line 10. Modify to "…have increased rapidly exceeding mean global growth."

Line 36. Modify to "Discharge of DBPs within the cooling…"

Line 40. Modify to "…regularly involve the discharge large volumes of water into the marine environment."

Line 41. Modify to "…and its decreased density means it is at the sea surface. Chemicals such as DBPs contained in cooling water are likely to spread laterally…".

Line 83. Modify to "…contributions to VSLSs, in the form of…"

Line 84. Modify to "…50 % of the global coastal cooling…"

Line 87. Modify to "…we show oceanic distributions"

Section 2.1 title. Recommend changing to "Estimation of DBP production in cooling water from East Asian power plants".

Line 96. Modify to "…the ocean provides an unlimited water supply."

Line 136. Modify to "…discharged with the cooling water."

Line 170. Modify to "…the impact that atmospheric bromoform abundances have upon on the flux calculations"

Line 185. Modify to "…bromoform for the three different emission strength scenarios with the Lagrangian…"

Line 187. "…(temperature, and winds)…"

Line 210. "…than the Ziska2013 emissions."

Line 218. "…and over another region from China…"

Line 219. "…we refer to this as the subtropical box…"

Line 221. "…in the region of interest…"

Line 222. "Non-volatile DBPs from cooling water usually accumulates…"

Line 229. "…in the South China Sea suggests only small contributions…"

Line 234. "Figure 3, because the volatile DBPs…"

Line 235. "…for the three emissions scenarios LOW…"

Line 236. "…smaller spread compared to the non-volatile DBPs."

Line 263. "…the Ziska2013 biogenic emissions are spread out…"

Line 264. "…similar total emissions as in the LOW emission…"

Line 283. "…the three scenarios…"

Line 292. "These differences are maximised…"

Line 319. "…in the tropical marine boundary layer where mixing ratios during DJF…".

Line 321. "…Ziska2013-Mixed that include…"

Line 323. "…the maritime continent, which increases tropical…"

Line 324. "…and even more so in the MODERATE run where…"

Line 327. "…that can lead to entrainment…"

Line 329. "…occur frequently in this region in both seasons…"

Line 352. "…and 17 pmol $L^{-1}$…"

Line 405. "…concentrations to be between…"

Line 410. "…in the form of anthropogenic…"

Line 413. "…in this region and might explain some…"

Line 423. "…emissions with only slightly less bromoform (0.15–0.16 ppt) being transported into the UTLS…"

Line 436. "Desalination is mostly done in the Arabian Peninsula…"

Line 443. "…areas (Maas et al., 2019), respectively."

---

## Referee Comment (RC1) · Anonymous Referee #1 · 29 Feb 2020

Maas et al. 2020 presents an estimate of near-coastal flux of by-product CHBr3 emissions from power plant discharges in Asia and its impact on atmospheric bromine loading. This analysis is based on some recent water sample measurements from power plant cooling water and surrounding waters, with the help of Lagrangian trajectory model calculation. This is an interesting study and provides some helpful information in terms of quantifying the anthropogenic contribution to the atmospheric bromine budget. However, I have several major concerns on the method, lack of adequate comparison for the most important region in this study (East Asia), and the major conclusion. These concerns should be addressed before the paper is considered for publication in ACP.

1. Section 2.3. The authors mention that the FLEXPART is run using the meteorological input stem from the ERA-Interim reanalysis . . . The FLEXPART simulations were performed for the boreal winter and summer seasons, for a total of three months with a one-month spin-up. Credible estimate of contribution of surface to UT/LS transport rely on the use of a model that can properly represent this transport process. At minimum, in the case no transport evaluation is conducted for this study, you need to provide adequate peer-reviewed results showing FLEXPART-based analysis is suitable for this study; that it is adequate in representing the surface to UT/LS transport within the Asian tropical/subtropical deep convection and the Asian summer Monsoon over the continent.

2. Second, please state clearly the year of DJF & JJA months you are using to drive the FLEXPART simulation. I am also not convinced a single year (with only 2-seasons) simulation is statistically adequate to quantify the transport from surface to the UT/LS in Asia. The authors need to decide an appropriate length (number of years) to address such transport using FLEXPART and provide a discussion on the year-to-year variability of the above transport. My suggestion is that at minimum you need a 10-year simulation to cover a few full cycles of QBO and ENSO, which have significant impacts on the dynamical transport relevant to this study.

3. This study is based extrapolating the information from a limited number of power plant effulent to the entire Asia power plants. As discussed in section 5, both the simulated oceanic and marine boundary layer concentrations of CHBr3 from this study, particularly those from the HIGH scenario, are larger than most of the previous observations in general. The regions that where 90% of the largest simulated concentrations (see Figures 5, 6, 7) display extremely high level of bromoform levels compared to the original Ziska 2013 results. Yet, no comparison with previous measurements were presented in this work. NASA has recently conducted an aircraft field campaign KORUS-AQ (https://www-air.larc.nasa.gov/missions/korus-aq/index.html) in this region with extensive airborne measurements of CHBr3 (from the Whole Air Samplers, PI Donald Blake) from surface to mid/upper troposphere that is highly relevant to this

study. These measurements are publicly available at https://www-air.larc.nasa.gov/cgi-bin/ArcView/korusaq#BLAKE.DONALD/. I strongly encourage the authors use the KORUS-AQ CHBr3 measurements to evaluated the simulated FLEXPART CHBr3 from the three scenarios for a proper assessment of the design of this experiment to see whether the extrapolation method used in this work is a reasonable approach.

4. Section 4.2: the discussion on the vertical transport of bromoform in the troposphere. While tropical deep convection plays an important role in vertical lofting near the EQ, vertical lofting in subtropical Asia and East Asia is primarily driven by the Asian Summer Monsoon in the summer season. These transport processes were not discussed adequately in this work and past literature were not referenced either. Please add.

5. Figures 8 and 9 and related discussions. Using a climatological cold point altitude of 17km for discussion of vertical lofting and entrance to LS is not adequate, and this is particularly not suitable for the subtropical box (Figure 8). The tropopause in this region is likely very different from the tropics and can be highly variable due to seasons or other dynamical processes. I would suggest the authors to use the tropopause height and potential temperature fields from ERA-Interim reanalysis. Only when the vertically lofted airmass cross the tropopause and enter beyond the 370-380K potential temperature, the amount of the remaining CHBr3 within the airmass would have a chance to survive the transport process, make to the stratosphere and have an impact on stratospheric bromine loading.

6. Final major comment on the main conclusion of this work. With all the previous potential issues I have noted above, the authors concluded that these anthropogenic emissions only contribute 0.02-0.03 ppt to the stratospheric bromine budget. I find it it not convincing, from the results presented in this work, to draw the conclusion that anthropogenic sources are important enough to be considered for future estimates of atmospheric bromine input. While local concentrations are high, due to the lack of efficient vertical delivering mechanism, these emissions have little chance reaching the

stratosphere. This has been the conventional understanding on the vertical transport efficiency of very-short-lived bromine species, and is seemingly confirmed again in this study.

Minor comments: 1. Lines 60-62. These statements are missing proper references. 2. Lines 149-152: Please list what are the non-volatile and volatile DBPs considered in this experiment 3. Figure 1. It would be helpful if you can add the locations of Table 1 results (the ones that in the region) on this plot, marked with a different symbol.

---

## Referee Comment (RC3) · Anonymous Referee #2 · 13 Mar 2020

I have another general comment on the paper. I think the authors should address this comment prior to publication.

I think it would really strengthen the paper if the authors included in the conclusions of the paper a comprehensive and thorough discussion of the uncertainties and limitations present in the work. As it is, it is very difficult for a reader to assess the different sources of error and uncertainty, and therefore to judge the authors claims and hypothesis. Following this, I think this would also allow a more precise identification of the required future work (this should be included as well) that needs to be undertaken to provide further proof/disproof of this hypothesis.

[Figure]

2020.

---

## Referee Comment (RC4) · Anonymous Referee #3 · 22 Mar 2020

The manuscript is an interesting manuscript that assess the amount of bromoform produced from power plant cooling water treatment in East and Southeast Asia. The spread of bromoform is simulated as passive particles that are adverted using the 3-dimensional velocity fields from the high-resolution ocean general circulation model. The manuscript is worth publication after minor revision.

Detail comment 1. Include full name of FLEXPART in the abstract. 2. What the author mean by "we expect" in their sentence "From comparison of our model results to observations, we expect initial bromoform concentrations between 20–60 $\mu$g L-1 used for the two lower scenarios, to be most realistic" in the abstract?. I think more proper word should be used. 3. Introduction, Line 39-40: Include reference. 4. Line 77-78: "Furthermore, new measurements of bromoform in disinfected cooling water have

become available suggesting potentially higher concentrations of up to 500 nmol L-1 (Yang, 2001)". Is there any latest reference to represent "new measurements"? 5. Line 128: The DRAKKAR Group, 2007: Is this a reference? If it is a reference, please list it in the Reference list.

———————————————————

---

## Author Response (AR1)

**Author response to "Simulations of anthropogenic bromoform indicate high emissions at the coast of East Asia"**

Josefine Maas[1], Susann Tegtmeier[1,*], Yue Jia[1,*], Birgit Quack[1], Jonathan V. Durgadoo[1] and Arne Biastoch[1,2]

[1]GEOMAR Helmholtz Centre for Ocean Research Kiel, Kiel, Germany
[2]Kiel University, Kiel, Germany
*now at: Institute of Space and Atmospheric Studies, University of Saskatchewan, Saskatoon, Canada

We thank the reviewers for their valuable comments and suggestions, which have helped us to significantly improve the paper in revision. We appreciate the effort it took the referees to help with their many suggestions and hope we can satisfactorily reply to the comments. In the following, we repeat the comments of all three referees in black, followed by our responses in blue. The changes made in the manuscript are highlighted and attached to this document.

**Referee 1**

Maas et al. 2020 presents an estimate of near-coastal flux of by-product CHBr3 emissions from power plant discharges in Asia and its impact on atmospheric bromine loading. This analysis is based on some recent water sample measurements from power plant cooling water and surrounding waters, with the help of Lagrangian trajectory model calculation. This is an interesting study and provides some helpful information in terms of quantifying the anthropogenic contribution to the atmospheric bromine budget. However, I have several major concerns on the method, lack of adequate comparison for the most important region in this study (East Asia), and the major conclusion. These concerns should be addressed before the paper is considered for publication in ACP.

> We thank the reviewer for his/her valuable comments which have helped us to improve the paper in revision. We have addressed the major comments by:
> - Including a more detailed description of the FLEXPART model and validation studies
> - Extending our one-year FLEXPART simulations to four years and including a discussion of interannual variability of VSLS transport from available literature
> - Including a comparison of our results to measurements from the KORUS-AQ campaign
> - Improving the discussion of bromoform transport in East and Southeast Asia
> - Reformulating the conclusions

1. Section 2.3. The authors mention that the FLEXPART is run using the meteorological input stem from the ERA-Interim reanalysis . . . The FLEXPART simulations were performed for the boreal winter and summer seasons, for a total of three months with a one-month spin-up. Credible estimate of contribution of surface to UT/LS transport rely on the use of a model that can

properly represent this transport process. At minimum, in the case no transport evaluation is conducted for this study, you need to provide adequate peer-reviewed results showing FLEXPART-based analysis is suitable for this study; that it is adequate in representing the surface to UT/LS transport within the Asian tropical/subtropical deep convection and the Asian summer Monsoon over the continent.

> Thanks for the comment. We have included a more detailed description of the FLEXPART model and added the relevant citation of the convection validation paper from Forster et al. (2007). We have also added reference studies that have successfully used FLEXPART to simulate transport pathways over the Indian Ocean, Maritime Continent and West Pacific (e.g., Fiehn et al., 2017; 2018; Fuhlbruegge et al., 2016; Marandino et al., 2013; Tegtmeier et al., 2013; 2020). As these studies have included model validation based on comparisons to available aircraft measurements, we use them here as references justifying the choice of FLEXPART for our simulations.

> l. 226 ff.: "The FLEXPART model includes parameterisation for moist convection (Forster et al., 2007) and turbulence in the boundary layer and free troposphere (Stohl and Thomson, 1999). It has been used in previous studies with a similar model setup and shown robust VSLS profiles compared to observations (e.g. Fiehn et al., 2017; Fuhlbrügge et al., 2016; Tegtmeier et al., 2020a)."

2. Second, please state clearly the year of DJF & JJA months you are using to drive the FLEXPART simulation. I am also not convinced a single year (with only 2-seasons) simulation is statistically adequate to quantify the transport from surface to the UT/LS in Asia. The authors need to decide an appropriate length (number of years) to address such transport using FLEXPART and provide a discussion on the year-to-year variability of the above transport. My suggestion is that at minimum you need a 10- year simulation to cover a few full cycles of QBO and ENSO, which have significant impacts on the dynamical transport relevant to this study.

> We agree that one single year is not sufficient to capture all variations of atmospheric transport processes. Therefore, additional FLEXPART simulations were performed using the same setup as the existing runs. We now include FLEXPART simulations for four years (2015-2018) for the boreal winter (December–February, DJF) and summer (June–August, JJA) seasons, respectively. Each run is conducted with a two months spin-up phase.
> We found relatively small interannual variations in our results and therefore decided to not include a 10-year time period. A full discussion of the impact of different atmospheric modes such as ENSO and Indian Ocean dipole can be found in Tegtmeier et al. (2020), who simulated CHBr$_3$ entrainment for a 35-year long time period and found interannual variations of up to 15 % to be much smaller than seasonal variations of up to 50 %. A short discussion of the impact of interannual transport variations has been added to the manuscript.

3. This study is based extrapolating the information from a limited number of power plant effluents to the entire Asia power plants. As discussed in section 5, both the simulated oceanic and marine boundary layer concentrations of CHBr3 from this study, particularly those from the

HIGH scenario, are larger than most of the previous observations in general. The regions that where 90% of the largest simulated concentrations (see Figures 5, 6, 7) display extremely high level of bromoform levels compared to the original Ziska 2013 results. Yet, no comparison with previous measurements were presented in this work. NASA has recently conducted an aircraft field campaign KORUS-AQ (https://www-air.larc.nasa.gov/missions/korus-aq/index.html) in this region with extensive airborne measurements of CHBr3 (from the Whole Air Samplers, PI Donald Blake) from surface to mid/upper troposphere that is highly relevant to this study.

These measurements are publicly available at https://www-air.larc.nasa.gov/cgi-bin/ArcView/korusaq#BLAKE.DONALD/. I strongly encourage the authors use the KORUS-AQ CHBr3 measurements to evaluate the simulated FLEXPART CHBr3 from the three scenarios for a proper assessment of the design of this experiment to see whether the extrapolation method used in this work is a reasonable approach.

Thanks for the suggestion. In addition to the already used observational data, we now include air measurements from the KORUS-AQ campaign over Korea in section 5.2. The comparison of the KORUS-AQ CHBr$_3$ measurements with the CHBr$_3$ obtained from FLEXPART simulations suggests very good agreement in this region for the MODERATE scenario. As existing bottom-up scenarios (Ziska et al., 2013) show significantly lower mixing ratios over South Korea and the Yellow Sea, the new comparison suggests that anthropogenic sources are required in order to explain observed CHBr$_3$ values. The comparison also confirms earlier conclusions that the HIGH scenario is unrealistic and results in too high abundances of atmospheric CHBr$_3$. We have revised the conclusion and discussion section taking the new information into account.

l. 445ff:" An extensive study of atmospheric measurements over South Korea and adjacent seas was performed in spring (May and June) 2016 from the Korea–United Sates Air Quality Study (KORUS-AQ; https://www-air.larc.nasa.gov/missions/korus-aq/). The aircraft measurements of various VSLSs including bromoform were repeatedly taken between 0 and 12 km in the region between 30° N-40° N and 120° E-145° E coinciding with our subtropical box discussed earlier (Figure S2). The data used here, is based on the 60 second merged dataset from all flight sections. In the campaign region around South Korea, an average bromoform atmospheric mixing ratio from all sections of 2.5±1.4 ppt was measured in the lower 100 m (Figure 8). In comparison, our simulations for the Ziska2013+MODERATE scenario show an average mixing ratio of 3.8±1.4 ppt in the lowest 100 m in the subtropical box during JJA. The simulations based on Ziska2013 give a bromoform mixing ratio of only 0.3±0.1 ppt for the same altitude range demonstrating that the additional anthropogenic bromoform sources results in a much better agreement with the observations in the marine boundary layer around South Korea.

Above the boundary layer, mixing ratios from KORUS-AQ rapidly decline to 0.5-0.7 ppt in the 3-9 km altitude range (**Fehler! Verweisquelle konnte nicht gefunden werden.**). Here, the Ziska2013+MODERATE simulation suggest seasonal mean mixing ratios between 0.4-0.7 ppt, which fit very well to the KORUS-AQ data. Simulations based on Ziska2013 suggest 0.2 ppt bromoform in this region clearly underestimating the observations (**Fehler! Verweisquelle konnte nicht gefunden werden.**). Between 9 and 12 km, the observed bromoform values drop sharply to values around 0.2±0.08 ppt suggesting that the airplane probed air masses above the convective

outflow. The smooth seasonal mean profiles from the two simulations do not show such sharp decrease of values and in consequence the lower Ziska2013 results agree better with the observations in the region 9-12 km. In general, the comparison with the KORUS-AQ data shows that simulations agree quite well the observations in the middle troposphere when anthropogenic emissions from cooling water treatment in East Asia are included based on the MODERATE scenario."

4. Section 4.2: the discussion on the vertical transport of bromoform in the troposphere. While tropical deep convection plays an important role in vertical lofting near the EQ, vertical lofting in subtropical Asia and East Asia is primarily driven by the Asian Summer Monsoon in the summer season. These transport processes were not discussed adequately in this work and past literature were not referenced either. Please add.

We have improved the discussion of the vertical transport of bromoform and how much of it is driven by tropical convection versus vertical lofting by the Asian Summer Monsoon. The impact of these processes on VSLS transport in the East and Southeast Asia region has been analysed and discussed by a number of recent studies (e.g., Fuhlbruegge et al., 2016; Hossaini et al., 2016; Fiehn et al., 2017) and we thus now include a summary of their most important findings in the introduction section. In addition, we have added past literature important for the overall discussion of tropospheric transport pathways in this region.

l. 80ff: "The Asian summer monsoon represents another important pathway to the lower stratosphere (e.g., Randel et al. 2010) entraining mostly Southeast Asian planetary boundary layer air. The monsoon also has the potential to include VSLSs emitted from the Indian Ocean and Bay of Bengal (Fiehn et al., 2017, 2018b). Model simulations suggest that the monsoon circulation transports the oceanic emissions towards India and the Bay of Bengal, from where they are convectively lifted and reach stratospheric levels in the south-eastern part of the Asian monsoon anticyclone. The stratospheric bromine injections from the tropical Indian Ocean and West Pacific depend critically on the seasonality and spatial distribution of the emissions (Fiehn et al., 2018a). Model studies based on bottom-up emission estimates indicate global bromoform maxima over India, the Bay of Bengal, and the Arabian Sea as well as over the Maritime Continent and West Pacific (Tegtmeier et al., 2020a). While aircraft measurements in the West Pacific have confirmed high concentrations of bromoform (Wales et al., 2018), the role of the Asian monsoon as an entrainment mechanism for VSLSs has not been confirmed yet due to the lack of observations in this region."

l. 409:" During the Asian summer monsoon, the region of main upward transport of VSLS lies at about 20°N over the Indian Ocean so that the main stratospheric injection region of VSLSs shifts to the Bay of Bengal and northern India (Fiehn et al., 2018b). However, most of the boundary layer bromoform from anthropogenic sources stays in the northern hemisphere around the coastline of China and over the West Pacific thus decoupled from the monsoon convection."

5. Figures 8 and 9 and related discussions. Using a climatological cold point altitude of 17km for discussion of vertical lofting and entrance to LS is not adequate, and this is particularly not suitable for the subtropical box (Figure 8). The tropopause in this region is likely very different

from the tropics and can be highly variable due to seasons or other dynamical processes. I would suggest the authors to use the tropopause height and potential temperature fields from ERA-Interim reanalysis. Only when the vertically lofted air mass cross the tropopause and enter beyond the 370-380 K potential temperature, the amount of the remaining $CHBr_3$ within the air mass would have a chance to survive the transport process, make to the stratosphere and have an impact on stratospheric bromine loading.

> We agree with this comment and have adapted our analyses. We now derive the $CHBr_3$ mixing ratios at the level of the ERA-Interim cold point tropopause taking into account temporal and spatial resolutions of this level. New versions of Figures 8, 9 and 10 show $CHBr_3$ at the cold point tropopause as derived from the ERA-Interim data (and not at 17 km as in the old version of the manuscript). As the cold point is the dehydration point of air masses on their way to the stratosphere, there will be very little impact of falling rain or ice on the $CHBr_3$ product gases above this level, which is therefore commonly used as the stratospheric injection level in VSLS studies. In the new version of the manuscript, we discuss the $CHBr_3$ contribution to the stratospheric bromine loading based on the mixing ratio at the cold point tropopause.

> l. 260f: "The UTLS region is calculated as the height of the cold point tropopause, which has been derived from ERA-Interim model level data at 6 hourly resolution (Tegtmeier et al., 2020b)."

6. Final major comment on the main conclusion of this work. With all the previous potential issues I have noted above, the authors concluded that these anthropogenic emissions only contribute 0.02-0.03 ppt to the stratospheric bromine budget. I find it not convincing, from the results presented in this work, to draw the conclusion that anthropogenic sources are important enough to be considered for future estimates of atmospheric bromine input. While local concentrations are high, due to the lack of efficient vertical delivering mechanism, these emissions have little chance reaching the stratosphere. This has been the conventional understanding on the vertical transport efficiency of very-short-lived bromine species and is seemingly confirmed again in this study.

> True enough, the amount of stratospheric halogen resulting from anthropogenic activities is rather small given that the majority of power plants is not located in the tropics. However, anthropogenic VSLS show high accumulations in the boundary layer and can change the tropospheric bromine budget and ozone chemistry. While we have not analysed this aspect in our current study, this should be investigated in follow-on projects. We have reformulated the conclusion to make it clear that we refer to the total bromine budget here and not to the impact of anthropogenic VSLS to the stratospheric bromine budget.

Minor comments:
1. Lines 60-62. These statements are missing proper references.
> We added the following references.

l. 62ff: "Top-down bromoform emission estimates, on the other hand, are based on global model simulations adjusted to match available aircraft observations (e.g. Butler et al., 2007; Liang et al., 2010; Ordóñez et al., 2012)."

2. Lines 149-152: Please list what are the non-volatile and volatile DBPs considered in this experiment

The non-volatile DBPs can be various. Thus, we mention bromoacetic acid as an example in paragraph 3 of section 2.2. The volatile DBP is explicitly given as bromoform in the text.

3. Figure 1. It would be helpful if you can add the locations of Table 1 results (the ones that in the region) on this plot, marked with a different symbol.

Only measurements from the South Korean power plant are situated in our study region. Therefore, we haven't added the locations to the plot as suggested but have added this information to the text (paragraph 2 of section 2.1).
l. 139: "Only Yang (2001) provides DBP measurements in East Asia."

**Referee 2**

This paper describes the estimation of sea surface concentrations of bromoform over East Asia resulting from the treatment of power station coolant water with chlorine-based disinfectants. The authors subsequently then estimate sea to air fluxes and test these within an atmospheric transport model to estimate the transport of bromoform in the atmosphere, in particular, via convection to the stratosphere. The authors estimate the sea surface concentrations using a bottom-up approach by first estimating bromoform within power station coolant waters, the discharge of this coolant water into the ocean, and then its subsequent transport in the ocean using oceanic transport model. Based on the bottom-up approach, the authors find a notable contribution to oceanic bromoform from this source that strongly affects oceanic bromoform concentrations close to the source region. Based on the sea-to-air fluxes and the atmospheric modelling, despite showing significant anthropogenic bromoform levels near the source regions and in the boundary layer, the authors find only a modest contribution of this source to stratospheric bromoform mixing ratios.

Bromoform has an important relevance to stratospheric ozone depletion. This paper therefore covers an important topic since current knowledge of bromoform sources is currently highly uncertain due to only limited monitoring of this gas. Attempts to improve our knowledge of its sources are therefore highly welcome. I therefore find that the paper sits well within the scope of the journal.

The subject matter and overall concept and methodology of the paper alone make it worthy for publication. On the whole, this is a good piece of work. Despite the manuscript's strengths though, there are several weaknesses in the work. I therefore have a list of general and specific

comments that will need to be addressed to improve the manuscript to a sufficient level to merit final publication.

General Comments

1. The overall framing of the paper. I think it would help ease some concerns (see for instance reviewer #1's comments) if the paper's main findings regarding anthropogenic bromoform were to be framed as a set of predictions (that can be tested) based on independent knowledge and data. Your bottom-up method makes a set of predictions based on the data you have used. Then you have made efforts to validate those predictions using sea surface bromoform concentration and atmospheric measurements; these data provide somewhat limited but promising support for your predictions. As it is, the manuscript abstract and conclusions contains various statements that are rather definitive, e.g., "We find that bromoform..." (line 26 in the abstract) and, similarly, "We find that..." (line 388 conclusions). When in fact the observational support appears promising yet far from definitive, and would require more dedicated observational monitoring to really prove this hypothesis. I therefore propose the authors modify the manuscript so it clearly follows this chain of reasoning: prediction/hypothesis (emission atmospheric, oceanic modelling) into observational support, followed by evaluation, and then lastly clear statements on what is needed to provide stronger support for this hypothesis, i.e., more specific and targeted observations.

> Thanks for this suggestion. We agree that rephrasing parts of the manuscript in the suggested manner would benefit the overall message, highlight the remaining (large) uncertainties and provide clear motivations for follow-on studies. We have rephrased the introduction and in particular the two last paragraphs of the introduction to present our analyses in the hypothesis-evaluation-conclusion framework. We have also rephrased parts of section 5 (observational support) and section 6 (discussion and conclusions) in a consistent manner.

2. Treatment/explanations of the UTLS and cold point. I think that the UTLS and cold point definition is too simplistic. The use of UTLS throughout does not capture any of the essential details of this complex atmospheric region. Sticking to a single altitude of 17 km for the cold point is also not really realistic when looking at different latitudes and seasons. Furthermore, sticking to the cold point as a definition of the lower stratosphere alone is also not entirely suitable and the suggestion from reviewer #1 to simply use the 380-390 K potential temperature line also misses some of the subtle complexity. I recommend that the authors consult Corti et al. 2005 and 2006 (see below) that both provide clear explanations and observational support for more nuanced explanations of dynamical interactions in the UTLS. First, the level of zero radiative heating (LZRH) is also useful a measure for whether air masses will undergo slow radiatively driven ascent to above the 380-390 K levels, and it is usually at 15 km for clear sky conditions. Second, they show that in-cloud (cirrus) radiative heating can be responsible for lofting cloud containing air masses from as low as 11 km upwards to eventually reach the stratosphere. I would recommend that the authors try to calculate the LZRH using the ECMWF meteorological fields to try to diagnose this to help determine which air masses at altitudes below 17 km are heading up or down; this would really strengthen the paper and strongly aid the interpretation of the results,

which is quite difficult at this point. If this is not possible it would be very useful to see at least the 11 and 15 km levels, but this would be a much weaker alternative.

> We agree with the reviewer comment that our treatment of the cold point as the 17 km level was too simplistic. We have changed the manuscript and now use the ERA-Interim cold point tropopause instead, taking into account its spatial and temporal variability. We have changed our Figures 8, 9 and 10 and have updated the discussion in our text, so that the stratospheric contribution is now based on the $CHBr_3$ values at the cold point level. We also agree with the reviewer that the UTLS is a complex region with various levels that are of importance for the vertical transport. It would indeed be interesting to see, which fraction of air masses that reaches the LZRH will also reach the cold point. However, such analyses of vertical transport characteristics within the TTL region is beyond the scope of this manuscript. Furthermore, we believe that for the VSLS and their soluble product gases the most important level is the cold point. By using the cold point as entrainment level, we automatically include all air masses that have crossed the LZRH and have undergone radiatively driven ascent as well as all air masses resulting from high-reaching convective detrainment.
>
> l. 260f: "The UTLS region is calculated as the height of the cold point tropopause, which has been derived from ERA-Interim model level data at 6 hourly resolution (Tegtmeier et al., 2020b)."

3. Many missing details. There are several missing important details from various sections of the paper, e.g., the year that is studied – this is not mentioned at all. I have addressed each of my concerns in specific comments below.

> Thanks for pointing out the missing details. We have added the details as suggested below including information on the years, which are studied.

4. Clarity of the manuscript. At many points the information given is insufficient to understand precisely what is being said. I have made various specific remarks below to help address this concern.

> We have rephrased the sections that are pointed out below in order to improve the clarity of the manuscript.

5. Duration of FLEXPART simulations/transport times to the stratosphere. The duration of the FLEXPART runs is 3 months. As shown in Corti et al. 2005/2006, transport times from the boundary layer to the 390 K level of 50 days, and so even with a 1 month spin-up, we can expect a delay of ~20 days for air masses from the beginning of the spin up to reach this level. Indeed, this appears to be visible in Figs. 8b and 9c and d. This complicates the interpretation of the results both for the 5-day averages (i.e., when are they in the course of the simulations), and for the time series in Figs. 8 and 9. Similarly, the bromoform emitted in the last ~50 days of the 3 month simulations has no chance to reach these altitudes. Please discuss these issues.

We have improved our simulations by allowing now for a two-month spin-up phase. As the bromoform lifetime estimates a range from 16 days at the ocean surface to 29 days in the TTL (Hossaini et al., 2012), we can expect the simulated bromoform to have reached a 'steady state' after two months. Transport acting on time scales longer than two months will not impact the atmospheric bromoform distribution given the short lifetime of the compound and can thus be neglected in our simulations.

We have changed our analysis further, which is now based on seasonal mean (JJA and DJF) bromoform mixing ratios averaged over 4 years and have updated Figures 8, 9 and 10 of the manuscript accordingly. As the seasonal mean averages at the cold point tropopause cannot contain significant amounts of bromoform emitted more than two months before the start of the season, the model setup with a two-month spin-up is justified for our analyses.

6. Discussion of key results. Despite being an important component of the bromine lofted to the stratosphere resulting from bromoform, the authors make no mention of product gases (PGs). This is particularly important in light of comment #2 above. I do not expect the authors to simulate PG formation, chemistry, washout, and transport, but it should be clearly explained that we expect much of the bromoform to be chemically processed into PGs during the 50 days or so of vertical ascent to 390 K from the boundary layer. Further on this point, it would be worth having some discussion on CTM studies showing the partitioning of bromoform and PGs at different levels in the atmosphere.

Thanks for the comment. We agree that PG entrainment is an important component of the VSLS contribution to stratospheric bromine and have added this aspect to Section 6 of our manuscript. Based on recent measurement campaigns (that can estimate a total PG entrainment from VSLS), modelling studies (that can distinguish between PG from bromoform and other VSLS) and our own results, we have added a discussion on how much additional PG entrainment could potentially be expected from anthropogenic bromoform sources.

l. 540ff:" This study focuses on source gas entrainment into the stratosphere and does not take into account additional product gas entrainment resulting from anthropogenic bromoform sources. Most observational and modelling studies estimate the total stratospheric bromine contribution to be split half and half into source and product gas contributions (Engel and Rigby, 2018 and references therein). Therefore, we estimate the total stratospheric bromine contribution in form of both, source gas and product gases, from the East and Southeast Asia anthropogenic bromoform sources to be around 0.24–0.30 ppt Br. Compared to a total stratospheric bromine contribution from all VSLSs of about 3–7 ppt Br (Engel and Rigby, 2018), the anthropogenic input estimated in this study provides only a minor contribution."

7. Year of study. As mentioned, the year of study is not mentioned in the paper. While it could be interesting to do a multi-year analysis, if this is beyond the capabilities/time constraints of the authors, an alternative would be to provide some climatological context on the specific year of

study. The WMO annual climate reports usually give a good region by region analysis that would help to set the meteorological context.

> Thanks for the comment. We have chosen year 2006 for the oceanic transport and have shown in Maas et al. (2019), that the interannual variations in the oceanic $CHBr_3$ transport are small.
> For the atmospheric analysis, additional FLEXPART simulations were performed using the same setup as the existing runs. We now include FLEXPART simulations for four years (2015-2018) for the boreal winter (December–February, DJF) and summer (June–August, JJA) seasons, respectively. As we know from existing studies that the interannual variations of $CHBr_3$ reaching the TTL of around 15 % are much smaller than the corresponding seasonal variations of around 50 % (Tegtmeier et al., 2020), we have decided to not include a long-term time series analysis. A short discussion of the impact of interannual transport variations has been added to the manuscript.

8. I think it would really strengthen the paper if the authors included in the conclusions of the paper a comprehensive and thorough discussion of the uncertainties and limitations present in the work. As it is, it is very difficult for a reader to assess the different sources of error and uncertainty, and therefore to judge the authors claims and hypothesis. Following this, I think this would also allow a more precise identification of the required future work (this should be included as well) that needs to be undertaken to provide further proof/disproof of this hypothesis.

> Thanks for the comment. Largest sources of uncertainty are the highly variable bromoform amounts found in chemically treated cooling water. We aim to take these uncertainties into account by including three different scenarios, which result in highly uncertain atmospheric concentrations. Based on comparisons with observations we are able to narrow the uncertainty range to the two lower scenarios. We include a more detailed discussion of these uncertainties and how they compare to other error sources in Section 6.

> l. 551:" Highest uncertainties in the estimates presented here, arise from the highly variable bromoform amounts found in chemically treated cooling water. Since there are very few and no recent measurements from power plants in East and Southeast Asia available, the chosen scenarios aim to give a range of environmental concentrations of anthropogenic bromoform. Additional uncertainties can arise from oceanic and atmospheric transport simulations and the parameterisation of air-sea fluxes. Since bromoform is emitted into the atmosphere on very short timescales, uncertainties arising from oceanic transport simulations are small compared to scenario uncertainties. Similarly, given the high saturation of anthropogenic bromoform in surface water, the sensitivity of our results to the air-sea flux parameterisation can be expected to be small. Atmospheric modelling can introduce additional uncertainties, especially regarding the contribution of anthropogenic sources to stratospheric bromine. VSLS FLEXPART simulations have been evaluated in numerous previous studies and shown in most cases good agreement with upper air observations (e.g., Fuhlbruegge et al., 2016; Tegtmeier et al., 2020a). In summary, uncertainties of our results are dominated by uncertainties of the bromoform concentrations in

undiluted cooling water. We have successfully reduced these uncertainties by nearly a factor of two based on comparing our predictions to available observations."

Line 43. It would make sense to show some of the chemical equations associated with bromoform formation in coolant water if they are known.

> We have added the following information to the manuscript 'The generally proposed mechanism for generating DBPs is the reaction of oxidants such as chlorine and ozone with organic and inorganic substances, such as bromide (Br-) and iodide (I-), in the water via the formation of hypobromous (HOBr) and hypoiodous (HOI) acid.' In addition, we provide some references that discuss the complex formation mechanisms more in detail.

The ordering of the introduction was a bit disjointed in my opinion and to also contain some text that is not relevant to the work at hand. I would remove the sentences between lines 50 and 55.

> We move parts of this section to the discussion.

In my opinion the text should be reordered such that the paragraph on lines 70-82 should be the second paragraph. The third paragraph should then be on lines 56-68. This would make a more logical flow in my opinion.

     done

Line 117. It was not clear what you meant by the settlement of pathogens. Do you mean growth?

     Changed to growth: "Colder water from mid- to high latitudes during winter requires less water treatment as the growth of pathogens takes longer compared to tropical or subtropical waters."

Line 137. I do not claim that this is important to their findings, but the authors should justify not including diffusion.

     We rephrased the sentence. Ariane generally is a purely kinematic tool. Without a diffusivity parameterisation, the calculations are fast and can be done for large spatial scale over long time periods, and many particles, which make particle density calculations quite robust.
     l. 168f: "The calculation of trajectories with Ariane is generally purely advective."

Line 140. Please mention the years you are looking at in this study and in Mass et al. 2019.

     We have added the year (see above).

Line 149. I could not make sense of the following text. It was not clear how point 2 relates to the text that follows or where point 2 is discussed. It was unclear what "distinguish" meant in this context – this is too vague and a more precise explanation would be welcome. Are points 1 and 2 meant to describe separate simulations? Separate processes? And why are 1 and 2 being treated separately at all? Clearer explanations here would be very helpful to the clarity of the manuscript.

     Thanks for the comment. We have rephrased the paragraph.

     l. 182ff: "We conduct two different simulations allowing us to analyse the spread of long-lived DBPs in general and the spread of bromoform as specific case. First, we simulate the spread of a passive tracer, which does not have any environmental sinks and represents any long-lived non-volatile DBP. We consider the full history of simulated particle positions, which is equivalent to assuming no particles getting lost through sinks in the ocean or emission into the atmosphere. The resulting distribution shows locations where non-volatile DBPs such as bromoacetic acid are transported through the ocean currents within one year.
     Second, we simulate the spread of bromoform as a major volatile DBP including the simulation of atmospheric fluxes and oceanic sinks. Each particle is assigned an initial mass of bromoform according to the amount of cooling water used by the respective power plant (**Fehler! Verweisquelle konnte nicht gefunden werden.**) and the bromoform concentration prescribed by

Line 171. The authors should make it clearer how the values of Ceq are calculated from the outgassed bromoform; this is currently not explained.

We added the equation for C$_{eq}$.
l. 200: "$C_{eq} = C_{air} \cdot H_{CHBr3}^{-1}$ (2)"

Lines 167-173. In general, this section of text needs to be clearer. This could be improved by stating that the low Ceq values are driven by low atmospheric vmr. It would also be clearer if the authors stated how Ceq relates to vmr.

This was done by adding the equation for C$_{eq}$.

Line 178. "Mean concentrations are calculated...". In air, CW, or Ceq or atmospheric vmr?

See answer below.

Line 178-179. "...of bromoform, characterised by the highest local concentrations, accumulate." This is not very clear.

We have rephrased the paragraph and tried to clarify the statistical approach.
l. 218ff: "Mean sea surface concentrations C$_w$ are calculated by averaging over the area where 90 % of all released bromoform accumulates. To this end, all grid cells are sorted according to descending bromoform concentrations and the average is calculated over the first grid cells that contain in total 90% of all bromoform. Maximum concentrations are calculated by averaging over the area where 10 % of the highest bromoform values accumulate."

Line 180. "Mean and maximum fluxes are calculated based on the same principle." What principle?
see answer above.

Line 181. "The annual mean atmospheric bromine input from industrial bromoform emissions". I think you mean resulting instead of "input".

Changed to: "The annual mean atmospheric bromine flux resulting from industrial bromoform emissions in East and Southeast Asia is derived from the air-sea flux maps of the whole domain."

Section2.3. We are missing a lot of details here. What resolution are the simulations carried out on? The same resolution as the meteorology? Are the emissions constant during a season? Are

Lagrangian particles emitted over the entire ocean and then the emission rate is proportional to the air-sea flux? What year are you looking at?

Thanks for the comment. We added the resolution of the simulations and further information.

l. 225: "Based on the seasonal mean emission maps, we obtain a source function of atmospheric bromoform. We simulate the atmospheric transport and distribution of bromoform for the three different emission strength scenarios with the Lagrangian particle dispersion model FLEXPART (Stohl et al., 2005). Seasonal mean bromoform emissions derived from the three scenarios are used as input data at the air-sea interface over the East and Southeast Asia area defined as our study region. The meteorological input data (temperature, and winds) stem from the ERA-Interim reanalysis (Dee et al., 2011) and are given on a 1°×1° horizontal grid, at 61 vertical model levels and a 3-hourly temporal resolution. The chemical decay of bromoform in the atmosphere was accounted for by prescribing a half-life of 17 days during all runs (Montzka and Reimann, 2010). The FLEXPART simulations were performed for boreal winter (December–February, DJF) and summer (June–August, JJA) seasons, respectively, each with a two-month spin-up phase, for the years 2015-2018. A total of 1000 particles are randomly seeded inside each grid box at each time step according to the air-sea flux strength."

Line 195 onwards. We are told that there are three additional runs that are made. Then, over the course of four paragraphs with at times unclear descriptions we are told details about them, but they are only referred to as 'first run' and then 'two additional runs', and then 'first of two runs'. These descriptions are imprecise and confusing. Please can the authors define three names for the runs in line 195 first and then describe them in the following text as "Run A does this....Run B does that ... etc".

We rephrased the paragraph clarifying the function of the different FLEXPART runs. We changed the names of the runs to make it clear which scenario they are based on and refer to each run by using the defined name throughout the rest of the manuscript.

l. 243: "We perform three additional FLEXPART runs, Ziska2013-EastAsia, Ziska2013 and Ziska2013+MODERATE based on the updated Ziska2013 emission inventory with the same FLEXPART configuration as described above for both seasons, DJF and JJA. As the Ziska2013 inventory currently presents our best knowledge of bottom-up derived bromoform emissions, it is of interest to analyse how much of these emissions can be explained by industrial sources and how much stems from natural sources.

The Ziska2013-EastAsia run uses only the Ziska2013 climatological emissions over the East and Southeast Asia area defined as our study region. Results from Ziska2013-EastAsia in the atmospheric boundary layer are used to compare the mixing ratios based on our anthropogenic emissions in the East and Southeast Asia region.

For comparisons of mixing ratios in the free troposphere and upper troposphere/lower stratosphere (UTLS), air-sea fluxes from other parts of the tropics also need to be taken into account as the time scales for horizontal transport are often shorter than the ones for vertical transport. Therefore, we set up the runs, Ziska2013 and Ziska2013+MODERATE. Ziska2013 uses the air-sea flux of the

Ziska2013 climatology for the global tropics and subtropics between 45° S and 45° N. As the Ziska2013 climatology is taking into account only very few northern hemispheric coastal data points, it likely neglects anthropogenic fluxes in some regions. Therefore, the Ziska2013+MODERATE run uses the Ziska2013 fluxes between 45° S and 45° N, but replaces them with the anthropogenic MODERATE flux values in all grid boxes where the MODERATE fluxes are larger than the Ziska2013 fluxes. The two runs, Ziska2013+MODERATE and Ziska2013, are used to evaluate the additional anthropogenic bromoform based on the MODERATE scenario in the UTLS region. The UTLS region is calculated as the height of the cold point tropopause, which has been derived from ERA-Interim model level data at 6 hourly resolution (Tegtmeier et al., 2020b)."

Line 214. The authors refer to means of the whole domain, but what is the domain?

This refers to the study area (90° E–165° E, 10° S–45° N). We have added the information to the text (l. 263).

Line 214-215. I could not understand the descriptions as they are for "Mean mixing ratios from the whole domain in the marine boundary layer and in the UTLS are given as the average over the 90 % area characterised by the highest local values, and maximum mixing ratios as the average over the largest 10 % (see Section 2.2)." Also, how did the authors decide upon the 90% and 10% levels?

The explanation for the statistical approach is given in section 2.2. For the atmospheric mixing ratios, we use the same analysis as for the oceanic concentration and air-sea flux. For averaging, we chose the area given by the 90% highest mixing ratios, as it includes the majority of anthropogenic $CHBr_3$ in this region and at the same time ensures that empty boxes or negligible small concentrations are not considered in the calculation of the mixing ratios. For similar reasons, we chose the area given by the 10% highest mixing ratios to derive maximum abundances ensuring that these estimates do not only depend on single local peaks.
We have added more information to explain our approach in section 2.2.

Line 216. The authors say they identify two regions. I think they mean define.

Changed the wording l. 265: "In a second step, we define two regions in order to analyse the vertical transport of bromoform into the free troposphere and into the UTLS."

Line 221. "...pattern in the research area of interest (Figure 3)." I think the authors mean region, and also which region? There are different areas being talked about. Please be precise for clarity.

Changed to l. 271: "The particle density distribution shows the annual mean DBP accumulation pattern in the region of interest in East and Southeast Asia (Figure 3)."

Line 231. Please can the authors show the Kuroshio current on the map?

We have added its approximate location to the text but decided against including one single current in our maps.

Lines 259-264. The section is unclear. The sentence on lines 262-264 is particularly unclear. Also, for clarity sake, please refer consistently to the Ziska et al. 2013 emissions as Ziska2013. These sentences are confusing because information is expressed imprecisely and there are references to prior statements that themselves unclear. Please try to arrange the information clearly, methodically, and logically.

Thanks for pointing this out. We have improved the clarity and message of this paragraph.

l. 314: "The annual bromine input from the ocean into the atmosphere in form of bromoform emissions in the East and Southeast Asia region is 118 Mmol Br according to the observation-based inventories from Ziska2013 (**Fehler! Verweisquelle konnte nicht gefunden werden.**). Our simulations suggest that the anthropogenic input alone amounts to 100, 300 and 500 Mmol Br a$^{-1}$ (LOW, MODERATE, HIGH) for the same region, which corresponds to almost 99 % of the bromine produced during cooling water treatment in the power plant for each scenario. This implies that all bromoform from cooling water treatment is eventually outgassed from the ocean into the atmosphere. While average and maximum air-sea fluxes of anthropogenic bromoform are much higher and confined to small areas around the discharge locations, the Ziska2013 air-sea fluxes are distributed along all coastlines and the equator and result in similar total annual mean Br flux as the LOW emission scenario (**Fehler! Verweisquelle konnte nicht gefunden werden.**). 90 % of the annual mean atmospheric bromine input from anthropogenic bromoform in East Asia occurs north of 20° N where 89–447 Mmol Br are released over one year, compared to the tropical Southeast Asian regions south of 20° N where only 10–52 Mmol Br a$^{-1}$ enter the atmosphere (from LOW to HIGH). In contrast, only 29 % of the total bromine from the Ziska2013 climatology in East Asia is released into the atmosphere north of 20° N, which suggests that the majority of the anthropogenic emissions from this region are missing in the Ziska2013 climatology."

Line 264. Is the implication of the results that most of the East Asian CHBr3 in Ziska2013 is anthropogenic in origin? I think the authors should state this more clearly if this is the prediction.

The results show, that a majority of the CHBr$_3$ that is released in East Asia is of anthropogenic origin, which however is largely missing in the Ziska2013 climatology. We rephrased the sentence to make this point clearer.

l. 324:"In contrast, only 29 % of the total bromine is released into the atmosphere north of 20° N from the Ziska2013 climatology, which suggests that the majority of the anthropogenic emissions from this region are missing in the Ziska2013 climatology."

Line 266. What is the 29% percentage relative to?
This refers to 29 % of the total bromine released into the atmosphere (see above).

Line 271. I found it odd that the authors make a 3 month long simulation and then only show a 5-day average in that entire simulation. Please can the authors explain or justify why such a short period of time is selected? Could the authors consider either monthly or 3-monthly averages as well? Also, which 5 days is this from within the 3 month simulation? All instances of this should be made clear and/or justified.

> We agree with this comment and have changed the configuration of the atmospheric simulations. The output is now presented as the seasonal average (section 2.3).

Lines 288-299. The authors discuss Figure 6 in relation to this text but do not mention the DJF results in Figure 7.

> We added the reference for Figure 7.
> "For both seasons, JJA and DJF, atmospheric bromoform based on industrial emissions is larger than atmospheric bromoform based on the Ziska2013 emissions (Figure 6d, Figure 7d)."

Lines 302-303. From the description given, it is not entirely clear what has been averaged. I assume it is a spatial average, but the authors should specify because the sentence implies it is spatial and temporal.

> We have added information on the averaging to the text.
> "In order to analyse atmospheric transport from the marine boundary layer into the free troposphere and UTLS, seasonal mean bromoform mixing rations are averaged over a subtropical box (30° N–40° N, 120° E–145° E, Figure 2) and a tropical box (10° S–20° N, 90° E–120° E, Figure 2**Fehler! Verweisquelle konnte nicht gefunden werden.**) […]."

Sentence on lines 321-324. I suggest placing this sentence prior to the sentence beginning "Thus, …" on line 320.
> done

Line 333. Please explain when and where the 5-day snapshot is.
> We changed the 5-day snapshot to a seasonal mean over several years.

Line 340 and 343. Please state when and where the vmr values are calculated for.

> We have added detailed information on when and where the mixing ratios are calculated for.

Lines 363-369. I am concerned here at the averaging approach reduces the complexity and is masking effects of over sampling of the open ocean regions. Thus, I am not sure this shows a good comparison of the same thing. I think this highlights that more thorough statistical analysis

needs to be carried out, i.e., a simple x versus y spatial scatter plot. Including this would strengthen the conclusions of the paper.

We agree that averaging can mask oceanic concentrations, especially at the coast. But we decided that it is not realistic to compare single point observational measurements with our large-scale modelling results. Especially, since the uncertainties in the modelling approach about the strength of the discharged $CHBr_3$ are very high. Therefore, we use the observations to assess, which of the scenarios chosen reproduces best the observational range of $CHBr_3$ in this region. We then conclude that from the three scenarios, the $CHBr_3$ emissions in the HIGH scenario are set too high and we expect industrial $CHBr_3$ emissions to be in the range of the LOW and MODERATE scenario.

Line 378. There is no mention of the year under comparison. Providing that there is overlap in the year, the KORUS-AQ data suggested by reviewer #1 could be useful here.

We chose the year 2016 for the FLEXPART simulation, which is the same year of the KORUS-AQ campaign. We added the information in the manuscript.

Line 388. Recommend changing "find" to predict.
done

Line 392. Make sure it is clear these are simulated vmrs.
done

Line 392. What is a cloud of high bromoform? Perhaps use something more precise like "A diffuse area with high bromoform abundances".
done

Line 395-396. Please be more specific as this sentence is unclear.
We rephrased the sentence to clarify the discrepancy between point measurements that do not capture the whole distribution of bromoform at the surface, and our simulation that includes also the highest concentrations directly at the coast and discharge locations.

Line 403. Recommend stating that the assumptions are reasonable in the majority of case since the cited observations show larger ranges than those stated here.

Good point, which we include in the discussion.

Line 406. Recommend stating that the HIGH results are only too high in the majority of cases.
done

Line 408. Recommend being more specific. Instead of "results" state bottom-up emissions, modelling, and observations.
done

Technical Comments

Recommendations. Please use a comma after uses of which in cases where it introduces a nonrestrictive phrase. When describing using a method from another publication use following instead of after.

> done

Line 10. Modify to "...have increased rapidly exceeding mean global growth."

> done

Line 36. Modify to "Discharge of DBPs within the cooling..."

> done

Line 40. Modify to "...regularly involve the discharge large volumes of water into the marine environment."

> Changed to: " …regularly involve the discharge of large water volumes into the marine environment."

Line 41. Modify to "...and its decreased density means it is at the sea surface. Chemicals such as DBPs contained in cooling water are likely to spread laterally...".

> done

Line 83. Modify to "...contributions to VSLSs, in the form of..."

> done

Line 84. Modify to "...50 % of the global coastal cooling..."

> done

Line 87. Modify to "...we show oceanic distributions"

> done

Section 2.1 title. Recommend changing to "Estimation of DBP production in cooling water from East Asian power plants".

> done

Line 96. Modify to "...the ocean provides an unlimited water supply."

> done

Line 136. Modify to "...discharged with the cooling water."

> nothing changed here

Line 170. Modify to "...the impact that atmospheric bromoform abundances have upon on the flux calculations"

> done

Line 185. Modify to "...bromoform for the three different emission strength scenarios with the Lagrangian..."

> done

Line 187. "...(temperature, and winds)..."

> done

Line 210. "...than the Ziska2013 emissions."

> done

Line 218. "...and over another region from China..."

> done

Line 219. "...we refer to this as the subtropical box..."

> done

Line 221. "...in the region of interest..."
    done
Line 222. "Non-volatile DBPs from cooling water usually accumulates..."
    "accumulate" refers to the non-volatile DBPs. We kept the phrase as is.
Line 229. "...in the South China Sea suggests only small contributions..."
    done
Line 234. "Figure 3, because the volatile DBPs..."
    done
Line 235. "...for the three emissions scenarios LOW..."
    Changed to: "…for the three cooling water discharge scenarios LOW…"
Line 236. "...smaller spread compared to the non-volatile DBPs."
    done
Line 263. "...the Ziska2013 biogenic emissions are spread out..."
    Changed to: "…the Ziska2013 air-sea flux is spread out…"
Line 264. "...similar total emissions as in the LOW emission..."
    Changed to: "…similar total fluxes as in the LOW emission scenario."
Line 283. "...the three scenarios..."
    nothing changed here
Line 292. "These differences are maximised..."
    done
Line 319. "...in the tropical marine boundary layer where mixing ratios during DJF..."
    done
Line 321. "...Ziska2013-Mixed that include..."
    done
Line 323. "...the maritime continent, which increases tropical..."
    done
Line 324. "...and even more so in the MODERATE run where..."
    done
Line 327. "...that can lead to entrainment..."
    done
Line 329. "...occur frequently in this region in both seasons..."
    done
Line 352. "...and 17 pmol L-1..."
    done
Line 405. "...concentrations to be between..."
    done
Line 410. "...in the form of anthropogenic..."
    done
Line 413. "...in this region and might explain some..."
    done
Line 423. "...emissions with only slightly less bromoform (0.15–0.16 ppt) being transported into the UTLS..."
    done
Line 436. "Desalination is mostly done in the Arabian Peninsula..."

Line 443. "…areas (Maas et al., 2019), respectively."

**Anonymous Referee #3**

The manuscript is an interesting manuscript that assess the amount of bromoform produced from power plant cooling water treatment in East and Southeast Asia. The spread of bromoform is simulated as passive particles that are adverted using the 3- dimensional velocity fields from the high-resolution ocean general circulation model. The manuscript is worth publication after minor revision.

Detailed comments

1. Include full name of FLEXPART in the abstract.

> done
> "Based on the emission estimates, atmospheric abundances of anthropogenic bromoform are derived from simulations with the Lagrangian particle dispersion model FLEXPART…"

2. What the author mean by "we expect" in their sentence "From comparison of our model results to observations, we expect initial bromoform concentrations between 20–60 µg L-1 used for the two lower scenarios, to be most realistic" in the abstract?. I think more proper word should be used.

> done
> "Comparing our model simulations with observations, the best agreement is achieved with initial bromoform concentrations in treated cooling water of 20-60 µg L$^{-1}$ used for the lower two scenarios."

3. Introduction, Line 39-40: Include reference.

> done
> "Cooling water effluents regularly involve the discharge of large water volumes into the marine environment (Khalanski and Jenner, 2012)."

4. Line 77- 78: "Furthermore, new measurements of bromoform in disinfected cooling water have become available suggesting potentially higher concentrations of up to 500 nmol L-1 (Yang, 2001)". Is there any latest reference to represent "new measurements"?

We added additional references, all of which are newer than the studies by Jenner et al. 1997.

"Furthermore, new measurements of bromoform in disinfected cooling water have become available suggesting potentially higher concentrations of up to 500 nmol L$^{-1}$ (Padhi et al., 2012; Rajamohan et al., 2007; Yang, 2001)."

5. Line 128: The DRAKKAR Group, 2007: Is this a reference? If it is a reference, please list it in the Reference list.

Thanks for notifying. We included the reference in the list.

[revised manuscript text omitted]

---

## Author Response (AR2)

**Author comment on Simulations of anthropogenic bromoform indicate high emissions at the coast of East Asia**

Josefine Maas[1,*], Susann Tegtmeier[1,2], Yue Jia[1,2], Birgit Quack[1], Jonathan V. Durgadoo[1,3] and Arne Biastoch[1,3]

[1]GEOMAR Helmholtz Centre for Ocean Research Kiel, Kiel, Germany
[2]Institute of Space and Atmospheric Studies, University of Saskatchewan, Saskatoon, Canada
[3]Kiel University, Kiel, Germany
*now at: Helmholtz-Zentrum Geesthacht, Institute of Coastal Research, Geesthacht, Germany

Correspondence to: Susann Tegtmeier (susann.tegtmeier@usask.ca)

Once more, we thank the reviewers for their input on the last manuscript version. The comments of both reviewers are listed below together with our responses (in blue). The marked-up version of the manuscript is attached. We hope that we could address all points satisfactorily.

**Anonymous referee #1**

The revision seems to have improved substantially from the previous version. Most of my concerns, as well as those from the other reviewers, from the originally submitted manuscript have been well addressed. The inclusion of multiple years of simulation and comparison with the KORUS-AQ have greatly strengthened the results. Overall, the revised manuscript looks good. My remaining major comment is about the Conclusions section. I understand that this section has been reworked substantially to address the concerns raised by me and reviewer #2. However, in present form, it becomes lengthy and to some point handwavy with not really relevant details. I agree with reviewer #2's suggestion that it is important to include discussions on uncertainties, various key processes that are important for source gas and product gas transport to assess Br budget. However, you should be mindful that conclusion and discussion need to be succinct and to the point, by focusing on the key messages. It would be good to hear what the other reviewers feel about the current conclusions, but, in my view, this section can use some significant condensation.

Thanks for the comment. We moved the paragraphs of uncertainties and product gas entrainment to different discussion chapters. Furthermore, we tried to shorten some paragraphs to better focus on the key message for the conclusion.

Some minor editorial comments:

P2 L72. It is not 100% accurate to say "Once bromoform is photolyzed …" as bromoform can also lost significantly due to reaction with OH. Suggest change to "Once bromoform is photochemically destroyed …"

Done.

P3 L87. Here and throughout the text: I think the more common term is "Western Pacific", not "West Pacific"

Done.

P3 L112. Bromine budgets -> bromine budget

Done.

P4 L146. I would suggest use either season-dependent or vary with season. "seasonally dependent" is probably not grammatically correct.

Changed to: "Vary with season."

P5 L156. Consider change to "To assess the long-term …"

Done.

P5 L161. Should be "1/12 degree"

Done.

P5 L168. Suggest change "generally purely advective" to "primarily advection"
Changed to: "primarily based on advection."

P5 L182. Add "a" before "specific"

Done.

P5 L183. Suggest change to "is representative of long-lived non-volatile DBP"

Done.

P6 L201. Should be "Henry's Law constant"

Done.

P7 L244. I would just use "bottom-up bromoform emissions", with no "derived"

Done.

P7 L247. Delete "area" after "Southeast". These are regions, not areas.

Done.

P7 L258. I don't think "evaluate the additional anthropogenic bromoform" is the correct description here. You are not really evaluating these results against observations (the KORUS-AQ comparison would be adequate to justify this point either as there were very few measurements up in the UT/LS region), but calculating the differences using sensitivity runs. I would suggest change to "quantify the additional anthropogenic bromoform contribution based on …"

Done.

P14 L514. I think here "substantial source of …" is adequate. The word "growing" is

misleading unless you have data suggesting the water chlorination consumption is growing, hence more formation of bromoform from anthropogenic sources.

Deleted "and growing" from the sentence.

P15 L528. Amounts -> amount

Done.

P15 L549. Budgets -> budget

Done.

P15 L549. Aspect -> aspects

Done.

P26 Table 2, last column. Please clarify whether it is ppt of CHBr3 or ppt of Br?

Done.

Figures 6-10 caption and color bar legend, and the corresponding discussion in the text. For clarity, please clarify whether these are ppt of CHBr3 or ppt of Br?

We adjusted the colorbar legend and the figure captions to clarify that they show ppt CHBr$_3$.

**Anonymous referee #2**

I thank the authors for addressing my comments. I think this is nearly ready now. I would like to see the following changes and then I think the paper will be ready for publication. There is still just one significant issue that needs to be solved related to the years of study.

Line 9. "Power plants in East and Southeast Asian economies… ". I think you need to be more specific. This could be read as the power plant size, number, or some other characteristic. You should make it clear that this is the number.

Done.

Line 12. Since this paragraph relates to the oceanic modelling, and the oceanic models support these findings, I would make it clear that "By means of Lagrangian analyses…" refers to oceanic "Lagrangian analyses"

Done.

The information on the years being studied is not mentioned in the abstract. I think this is fairly critical information given the strong arguments linking rapid economic development to increases in emissions. Furthermore, I suggest you link the concluding statement in the abstract (line 35) to the period over which this conclusion is valid.

Done. We added information about the model years from the OGCM and ERA-Interim in the abstract. Additionally, we added that the results refer to the current emissions of industrial

bromoform.

Line 19. Suggest making it clearer that these are concentrations in sea water. Perhaps use "in coastal waters" instead of "along the coastlines".

Done.

"Atmospheric and oceanic measurements cannot distinguish between naturally and industrially produced bromoform and all the top-down and bottom-up emission estimates discussed above automatically include the latter." I would just be a bit careful here. I think this is an over-simplification. For example, the top-down emissions from Ordóñez et al., 2012 actually prescribed a spatial distribution over the open ocean according to the satellite derived maps of chlorophyll-a, so this statement is not fully correct in this case. Perhaps lessen this to "…emission estimates discussed above potentially include the latter already."

Done.

Line 102. "Especially" redundant.

Done.

Section 2.2 and 2.3. I think the authors need to be a bit clearer on some of the assumptions and need to give more justification for some of the methodological choices. There is an apparent mismatch between the years run for the oceanic model (2005-2006) and the years run for the atmospheric model (2015-2018). A more logical choice would have been to use the same time period for each model, but perhaps this was not possible. This is not necessarily a problem, but I would like the authors to state clearly that such a mismatch exists, explain why it exists (I assume the oceanic forcing is not available for the 2015-2018 period?), to give at least some kind of justification (limited year-to-year variability?), and to explain what impact this has on their conclusions. This seems to me to be the last major issue requiring attention prior to publication. It would be useful to discuss the effects of this choice in the uncertainties discussion in Sect 6.

Added in L. 172: "The year chosen is the same as in Maas et al. (2019), where it is shown that interannual variability of surface velocity in the study region is small compared to seasonal variability."

Line 281. Change …is… to …are.
Nothing changed here. "[…] a relatively large amount of DBPs is transported eastward with the Kuroshio Current east of Japan into the North Pacific."

Line 284. Change the first DBPs to DBP.
Done.

Line 312. Change to "in the form"
Done.

Line 359. "… to the stratospheric halogen budget."

Done.

Line 368. "…compared to bromoform from climatological…"'
Done.

Line 403. "and up to the cold point"
Done.

Section 5.2. Can the authors please tabulate the information presented in the latter paragraphs where the various comparisons between the observed and simulated atmospheric CHBr3 mixing ratios are reported. Try make a table with the mixing ratios from Low, Moderate, and High for the different regions where there are observations and list the refs in the table perhaps by using asterisks. This would make it easier to absorb this information. In some places in the text it is pretty difficult to follow, and at one point the authors tell us that the observations from the Pearl River Delta compare well with the observations, but we are not told what the values are.
We revised the paragraph to better present the observational results in comparison with the modelled values of $CHBr_3$. We also added a table for a better overview.

Sect. 5.2 Can the authors please try to explain why the observations in the Strait of Malacca are much higher than in their simulation?
We condensed the comparison with observations to results from measurements around Singapore, which are better comparable to the modelled results.

Line 511. I think you should state that we also need dedicated monitoring at coastal sites near to these anthropogenic sources.
Done.

Line 514. Provides
Done.

Line 537. input of 0.12-0.15 ppt Br.
Done.

Line 542. in the form of both,
Done.

Line 550. The highest uncertainties
Done.

Line 559. shown in most cases to give good agreement
Done.

Figure 10. Please make the coastlines more visible. Perhaps black would be a more suitable colour.
We adjusted the figures to make the coastline more visible.

[revised manuscript text omitted]